# Improving Model Representation and Reducing KV Cache via Skip Connections with First Value Heads

**Zhoutong Wu**[1], **Yuan Zhang**[1], **Yiming Dong**[2], **Chenheng Zhang**[2], **Cong Fang**[2,3],

**Kun Yuan**[1,3,4][†], **Zhouchen Lin**[2,5,6][†]
{ztwu, zy1002, chenhengz}@stu.pku.edu.cn,
{fangcong, kunyuan, zlin}@pku.edu.cn, yimingdong_ml@outlook.com

[1] Academy for Advanced Interdisciplinary Studies, Peking University
[2] State Key Lab of General AI, School of Intelligence Science and Technology, Peking University
[3] AI for Science Institute, Beijing, China
[4] National Engineering Laboratory for Big Data Analytics and Applications
[5] Institute for Artificial Intelligence, Peking University
[6] Pazhou Laboratory (Huangpu), Guangzhou, Guangdong, China

## Abstract

Transformer models have driven breakthroughs across various language tasks by their strong capability to learn rich contextual representations. Scaling them to improve representation, however, often demands substantial memory and compute costs, such as the Key-Value (KV) cache used during auto-regressive decoding. Skip connections offer a promising way to improve representation without bloating resource usage, yet most prior works either improve expressivity while leaving KV costs unchanged, or reduce memory at the cost of weaker representation. In this work, we propose SkipV1Former, a Transformer variant that uses skip connections from the first layer's Value heads to strengthen model representation and reduce KV cache. Specifically, from the second block onward, each layer reuses half of its Value heads from the very first layer, while computing the other half as usual-cutting Value projections and V cache by nearly 50 %. Theoretically, we show that routing uncompressed first-layer Values into deeper layers restores information lost to compression and accelerates the model's implicit mesa-optimization-a key pattern of Transformer in auto-regressive tasks. Empirically, across different model scales, SkipV1Former delivers consistent reductions of approximately 25 % in KV cache while improving perplexity relative to standard Multi-Head Attention (MHA) Transformers and some advanced variants. Moreover, we propose a recipe for uptraining existing MHA Transformer checkpoints to SkipV1Former with only 10-15% additional compute. Finally, SkipV1Former can seamlessly combine advanced methods like Group-Query Attention and Multi-Latent Attention to achieve further KV cache savings and performance improvement. When combined with YOCO, it cuts KV cache size by nearly 50 % while still improving performance. The code is available at: `https://github.com/Zhoutong-Wu/SkipV1Former`.

## 1  Introduction

Large Language Models (LLMs) built on Transformer architectures [1] have achieved impressive performance in a wide range of language tasks such as dialogue generation and complex reasoning.

---

[†]Corresponding author.

39th Conference on Neural Information Processing Systems (NeurIPS 2025).

At their core, multi-head attention (MHA) [1] endows these models with a powerful ability to capture rich contextual representations. While modern models are scaling to billions or trillions of parameters [2, 3, 4] in pursuit of ever-stronger expressivity, their memory and computational demands grow excessively large, limiting practical deployment. In particular, the need to cache large Key and Value (KV) tensors in auto-regressive decoding presents a notable challenge for GPU memory [5].

Moreover, recent empirical studies suggest that beyond a certain scale, simply increasing model size yields diminishing returns in downstream performance [6]. As a result, a growing body of works is focusing on methods that boost model performance without a corresponding surge in resource consumption. One promising technique is (cross-layer) skip connections, which route features between non-adjacent layers to promote reuse. Existing skip-connection techniques in the Transformer family mainly fall into two categories:

1. Feature-augmentation approaches: these methods concatenate or linearly fuse representations from earlier layers into later layers [7, 8], analogous to DenseNet [9] or U-Net [10] designs in convolutional networks. By reusing feature maps, these skip connections bolster model representation and improve performance. However, they fail to reduce KV cache footprint compared to a standard Transformer of the same width and depth.

2. Layer-replacement approaches: these techniques selectively replace portions of a layer's Key or Value (or other components) with those drawn from other layers [11, 12]. They can reduce the KV parameter counts and thus the KV caching consumption. However, naively substituting layer outputs often leads to a drop in model quality, as it often weakens the model representation.

This raises a natural question: Can we design a skip-connection scheme that simultaneously enriches representations and reduces KV-cache/storage requirements? To achieve both goals simultaneously, one must decide which parameters are relatively redundant to be replaced, while identifying which intermediate features are critical to reuse for expressivity. In fact, by the No Free Lunch theorem [13], there is no universal recipe for shrinking parameter counts and boosting representation across all tasks. Such dual gains must exploit both the architecture and the task characteristics.

To address the problem, we propose SkipV1Former (skip connection with Value-1), a simple yet effective MHA Transformer variant that strategically integrates both objectives. In an $L$-layer SkipV1Former, it replaces half of each layer's Value heads (layer 2 to $L$) with the corresponding heads from layer 1. This design cuts Value cache size and the trainable Value projection parameters by nearly 50%, while preserving the total head count and architectural depth.

Based on the characteristics of auto-regressive tasks, we provide a theoretical analysis of the expressivity of SkipV1Former by viewing Transformers as mesa-optimizer [14], i.e., self-attention block can be interpreted as performing an optimization step on a latent objective function. We show that feeding uncompressed first-layer Value signals to deeper blocks via skip connections restores lost information and accelerates the mesa-optimization process.

Experiments on GPT-2 models demonstrate that SkipV1Former consistently lowers perplexity compared to standard MHA Transformers and even matches or outperforms advanced feature-augmentation methods—with $\sim$25 % less KV cache. Under the same cache-saving regimes, it significantly outperforms layer-replacement approaches like YOCO [15] and CLA [16]. Scaling to the 3B LLaMA model, SkipV1Former maintains improved perplexity and KV-cache efficiency versus the vanilla MHA Transformer.

Moreover, we propose an uptraining method to train a SkipV1Former from the checkpoint of an MHA Transformer, using only around 10 - 15% of original pre-training compute. Finally, SkipV1Former can be seamlessly combined with other advanced KV-saving approaches like Group-Query Attention and Multi-Latent Attention to attain even greater savings in KV cache and performance boost. By combining the Key reusing mechanisms in YOCO, we further propose a variant of SkipV1Former that saves $\sim$ 50% KV cache while improving model performance.

## 2 Related Works

**Skip Connections in Transformers.** Skip connections, which route the output of one layer directly into deeper layers or sublayers, are utilized in the original Transformer for stabilizing training [1].

Subsequent Transformer variants have leveraged skip connections to enrich representations by reusing and propagating low-level features into higher layers [7, 17, 18, 8, 19, 20], leading to improved performance. Some recent works further generalize skip connections to "hyper" connections that also fuse connections from width [19]. Another main thread leverages skip connections to reduce inference costs by parameter sharing or substituting [16, 15, 21]. In particular, sharing Keys and Values across layers yields a straightforward and effective KV-cache reduction [16, 15]. Other approaches approximate attention scores for similar efficiency gains [22]. In addition, skip connections have also been applied to combine Pre-LN and Post-LN advantages [23] or mitigate over-smoothing problem [24]. Among the above, the most related to our approach is ResFormer [25], which linearly adds the first layer's Value to every subsequent layer's Value, acquiring lower perplexity but without theoretical grounding or KV cache savings. Our SkipV1Former simultaneously enhances expressivity and compresses KV cache, backed by both theoretical analysis and empirical validation.

**KV Cache Efficient Methods [26].** Key-Value (KV) cache, which stores the intermediate attention Keys and Values during auto-regressive generation to avoid redundant computation, often dominates GPU memory usage. To alleviate this, a range of compression techniques have been proposed, including KV quantization [27, 28], efficient attention [29, 30, 31], sparsity methods [32, 33, 34, 35], and KV sharing [36, 11, 37, 38, 39]. Our approach falls into the KV sharing category, reducing the number of cache entries via feature reuse and substitution. Representative KV sharing methods include Multi-Query Attention (MQA) [36] and Group Query Attention (GQA) [11], which share the Keys and Values among all or a subset of heads in the same layer. Other works explore cross-layer reuse—recycling KV states from adjacent layers—such as YOCO [15], Cross-Layer Attention (CLA) [16], and Layer-Condensed KV (LCKV) [21]. Some works also attempt to share latent cache across different layers [40]. SkipV1Former is the first to reuse half of every layer's Value heads directly from Value-1, which boosts model performance using less parameters.

**Transformer as Mesa-Optimizer.** In studies on in-context learning of Transformers, recent works reveal that learned Transformers exhibit a "copying" behavior, whereby they aggregate or replicate context tokens into one representation [14], then implicitly perform optimization steps—akin to gradient descent [41, 42] or Newton's method [43]—on a context-dependent loss. Subsequent work has validated this meta-optimization view across a range of sequence-modeling tasks [14, 44]. Building on the mesa-optimizer viewpoint, we analyze SkipV1Former and show that its cross-layer Value skip effectively accelerates implicit optimization with uncompressed first-layer signals.

# 3 SkipV1Former

## 3.1 Background and Notations

We begin by reviewing the architecture of multi-head attention (MHA) Transformer. Let $X \in \mathbb{R}^{d \times n}$ be the input to a Transformer block, where $d$ is the embedding dimension and $n$ is the sequence length. Neglecting layer normalization for clarity, a standard $H$-head self-attention sublayer followed by a feed-forward network (FFN) is

$$Q^h = W_Q^h X, \quad K^h = W_K^h X, \quad V^h = W_V^h X, \quad h = 1, \cdots, H,$$

$$\text{Attn}(X) = X + \sum_{h=1}^{H} W_O^h V^h \text{softmax}\big((K^h)^\top Q^h\big),$$

$$\text{FFN}(X) = X + W_2 \, \text{ReLU}(W_1 X),$$

where $W_O^h \in \mathbb{R}^{d \times d_H}, W_Q^h, W_K^h, W_V^h \in \mathbb{R}^{d_H \times d}, W_1 \in \mathbb{R}^{r \times d}, W_2 \in \mathbb{R}^{d \times r}$, $d_H$ is the dimension of each head and $r$ is the dimension of the hidden layer. Stacking $L$ such blocks yields the full Transformer architecture. We focus on decoder-only Transformers in this work.

## 3.2 Our Methods

**Architecture.** We now introduce our SkipV1Former. In each layer $i = 2, \cdots, L$, SkipV1Former interleaves half of that layer's Value heads with the corresponding heads from layer 1, while computing the remaining parts in the standard MHA manner. Formally, letting $H' = H/2$, the attention in block

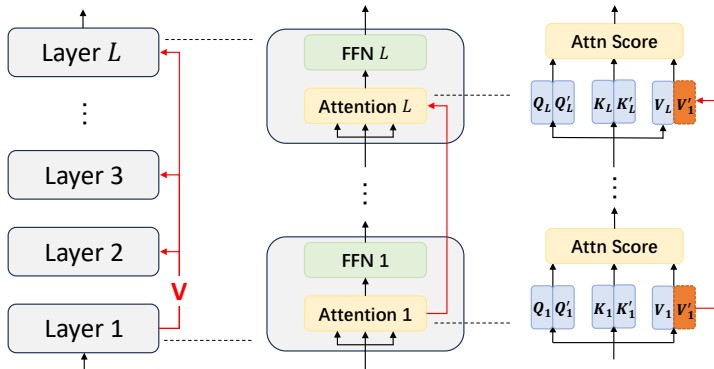

Figure 1: Overview of the SkipV1Former architecture. **Left:** A full $L$-layer decoder, where each layer from 2 to $L$ interleaves half of its Value heads with the corresponding heads from layer 1. **Center:** A zoomed-in view of the first and $L$-th layers, showing both the attention and the FFN sublayers. **Right:** Detailed illustration of the cross-layer Value skip connection, with orange shading indicating the first-layer Value heads that are injected into subsequent layers' head sets.

$i$ becomes

$$\text{Attn}(X) = X + \sum_{h=1}^{H'} W_O^h V^h \text{softmax}\big((K^h)^\top Q^h\big) + \sum_{h=H'+1}^{H} W_O^h \mathbf{V}_1^h \text{softmax}\big((K^h)^\top Q^h\big),$$

where $\mathbf{V}_1^h$ refers to the value of the $h$-th head of the first layer. Figure 1 gives an illustration of the architecture of SkipV1Former. Specifically, we use a fixed deterministic selection strategy: for layers 2-$L$ we always keep heads $0 \cdots H'$ and concatenate them with heads $H' + 1 \ldots H$ from layer 1. This continuity ensures a stable and aligned head ordering across layers, which is more hardware-friendly compared to other discontinuous methods [45] such as random selection. We also compare other head injection strategies in Appendix C.4.

Since half of the heads in layers $2 - L$ are drawn from layer 1, SkipV1Former cuts the number of $W_V$ parameters and Value cache by about 50% (i.e., 25% KV cache) without changing the total head count $H$ or layer depth $L$. We further propose a variant of SkipV1Former that utilizes the Keys sharing mechanism of YOCO (You Only Cache Once) [15] in Appendix C.5, which saves another 25% KV cache on top of SkipV1Former (50% KV cache in total). We focus this paper on SkipV1Former and leave the YOCO-based variant to Appendix C.5.

**Relations to Previous Methods.** Conceptually, SkipV1Former can be seen as an extension of ResFormer [25], which blends every layer's Value via a scalar interpolation, i.e.,

$$\text{Attn}(X) = X + \sum_{h=1}^{H} W_O^h \big(\lambda V^h + (1 - \lambda)V_1^h\big) \text{softmax}\big((K^h)^\top Q^h\big)$$

for some $\lambda \in \mathbb{R}$. In contrast, SkipV1Former inserts $H'$ first-layer heads directly into each deeper layers, thereby saving nearly 25% of KV cache whereas ResFormer cannot. In Section 4, we provide a theoretical analysis which not only motivates the specific design of SkipV1Former, but also offers insights into why ResFormer's interpolation with first Value benefits model performance.

## 4  Why Skip Connection with First Value Head Helps

While it seems natural to view SkipV1Former as more expressive than a variant that only reduces half the heads without injecting first-layer Values, it is shown that skipping from the first layer's Value yields notably better performance than skipping from later layers (see Section 6.5). This suggests that the first layer is uniquely informative for information propagation.

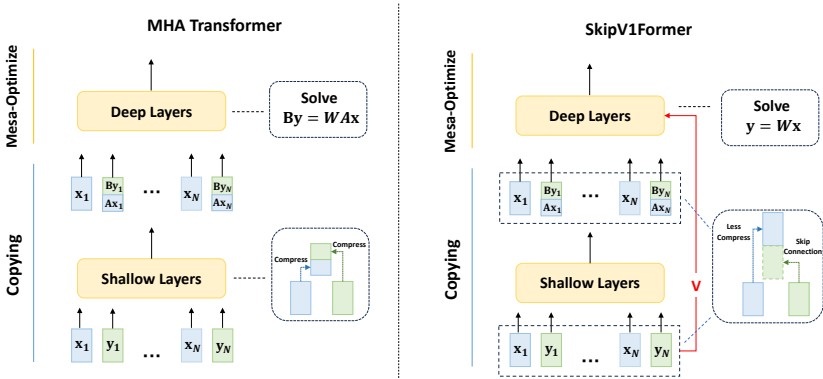

Figure 2: Standard MHA Transformer (left) compresses token pairs across layers, incurring information loss in mesa-optimization, whereas SkipV1Former (right) reintroduces raw token values into deep layers, preserving information fidelity.

A potential clue lies in the "copying mechanism" studied in in-context learning (ICL) [46]: early Transformer layers often act to bind multiple adjacent tokens into compact representations [14, 47]. This effect is especially prominent in shallow layers and underlies many ICL models' assumptions. For example, it is commonly assumed that input tokens take the form $\mathbf{e}_j = (\mathbf{x}_j, \mathbf{y}_j) \in \mathbb{R}^{d_x + d_y}$, where $\mathbf{x}_j \in \mathbb{R}^{d_x}$ and $\mathbf{y}_j \in \mathbb{R}^{d_y}$ are the sample and label of a training pair. Under this formulation, Oswald et al. [41] show that, for a linear regression problem, i.e., $\mathbf{y} = W \mathbf{x} + \varepsilon$, a single linear attention layer approximates a one-step gradient descent:

$$\mathbf{e}_j \leftarrow (\mathbf{x}_j, \ \mathbf{y}_j - \eta \nabla L(W) \mathbf{x}_j), \quad L(W) = \frac{1}{2N} \sum_{i=1}^{N} \|W \mathbf{x}_i - \mathbf{y}_i\|^2. \tag{1}$$

That is, the layer updates the token as if performing one step of gradient descent on $W$ to minimize the regression loss. Subsequent works [48, 49, 42] extend this mesa-optimization view to deeper and nonlinear settings [48, 49, 42].

However, a key assumption in this analysis is that each token *fully* encodes its $(\mathbf{x}, \mathbf{y})$ pair—a condition difficult to satisfy in practice due to limited embedding dimensionality. Compressing multiple tokens into one inevitably incurs information loss, which can lead to degraded quality of optimization updates in deeper layers.

SkipV1Former mitigates this by explicitly routing the first layer's Value into every deeper layer. Since the first layer processes the raw input tokens directly, its Value retain higher-fidelity representations of the sample and label. The skip connection with first-layer Value allows the model to restore much of the information lost through copying whereas existing residual connections cannot. This preserves more of the causal structure between samples and labels, which in turn enhances the model's reasoning capabilities. The overall intuition is illustrated in Figure 2. Moreover, prior works [48] show that mesa-optimization gradients in Transformers primarily flow through the Value pathway. This supports our design choice to interleave deeper layers' Value with the first-layer Value.

We further formalize this intuition on a simplified model to derive concrete performance improvement. Consider the linear regression problem in Eq. (1), where $\mathbf{x}_i, \mathbf{y}_i \in \mathbb{R}^d$ and $W \sim \mathcal{N}(0, I_d^2)$. Assume that the embedding dimension is $d$, so there will be information loss in the compression for MHA Transformer. For the model, consider a two-layer Attention-only Transformer with two heads per layer and linear attention. Denote by $L_1$ and $L_2$ the expected squared-error loss on predicting $\mathbf{y}_{n+1}$ of MHA Transformer and SkipV1Former, respectively. We show that under this setup:

**Theorem 1** (Informal)**.** *Assume that the first layer performs the copying mechanism, then there exists an independent constant $c > 0$, such that*

$$\min_{W_Q^h, W_K^h, W_V^h, H} L_2(\{W_Q^h, W_K^h, W_V^h\}, H) \leq \min_{W_Q^h, W_K^h, W_V^h, H} L_1(\{W_Q^h, W_K^h, W_V^h\}, H) - c.$$

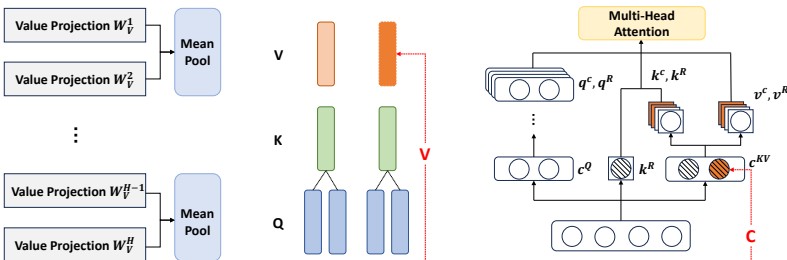

Figure 3: **Left**: Uptraining checkpoint conversion: we apply mean pooling over every two $W_V$ heads to reduce dimensional mismatch. **Middle**: Integration of SkipV1Former with GQA. **Right**: Integration of SkipV1Former with MLA; shaded regions follow the style of [50] and indicate cached components.

A formal statement and proof of the theorem are provided in Appendix A. Theorem 1 shows that SkipV1Former can drive the prediction error at least $c$ lower than MHA Transformer on this task. In the proof, we show that SkipV1Former can realize an "uncompressed" GD—just as in the ideal one-step update in Eq. (1). In this sense, SkipV1Former performs equivalently as an MHA Transformer with extra embedding dimension that can fully encode the $(\mathbf{x}, \mathbf{y})$ pair. Further discussions are provided in Appendix A.

# 5    Extensions & Practical Recipes

## 5.1    Uptraining from Pretrained MHA Checkpoints

Modern large language models are typically released with standard MHA Transformer checkpoints. To train a SkipV1Former from an MHA Transformer checkpoint, we follow a similar uptraining strategy to that of Ainslie et al. [11] for Multi-Query Attention [36]. Concretely, 1) Convert an MHA Transformer checkpoint into a SkipV1Former checkpoint by inserting a mean pool across every two head projections in layers beyond the first. 2) Initialize the new SkipV1Former layers with these pooled weights while preserving the original first-layer Value projections and continue pretraining. An illustration is shown in Figure 3. It is shown in Section 6.4 that this uptraining can match the perplexity of training from scratch using an additional 10%-15% of the original compute budget.

## 5.2    Combining with KV-Cache-Efficient Methods

Our first-value-head skip connection is compatible with most existing KV cache compression methods, such as sparsity methods and same-layer KV sharing methods. In this Section, we consider Group-Query Attention (GQA) [11] and Multi-Latent Attention (MLA) [50]. We also consider YOCO in Appendix C.5.

- **Group-Query Attention**: GQA divides query heads into groups, each sharing a single key and value head. To integrate, we first replace half of the Value heads with those from the first layer before grouping. Each group thus shares either a current-layer or first-layer Value head, enabling shallow-layer reuse without disrupting GQA semantics.

- **Multi-Latent Attention**: MLA uses a low-rank joint compression $\mathbf{c}$ for attention keys and values. To integrate with our technique, we interleave half of each layer's $\mathbf{c}$-dimensional latent vector with the corresponding first-layer latent features, reducing the $\mathbf{c}$-cache footprint.

See Figure 3 for an illustration. It is shown in Section 6.4 that combining our methods with GQA or MLA leads to another 25% or 50% reduction on KV cache and attaining lower perplexity. We also experiment on TinyLlama-1.1B model in Appendix B.2 and observe similar improvements.

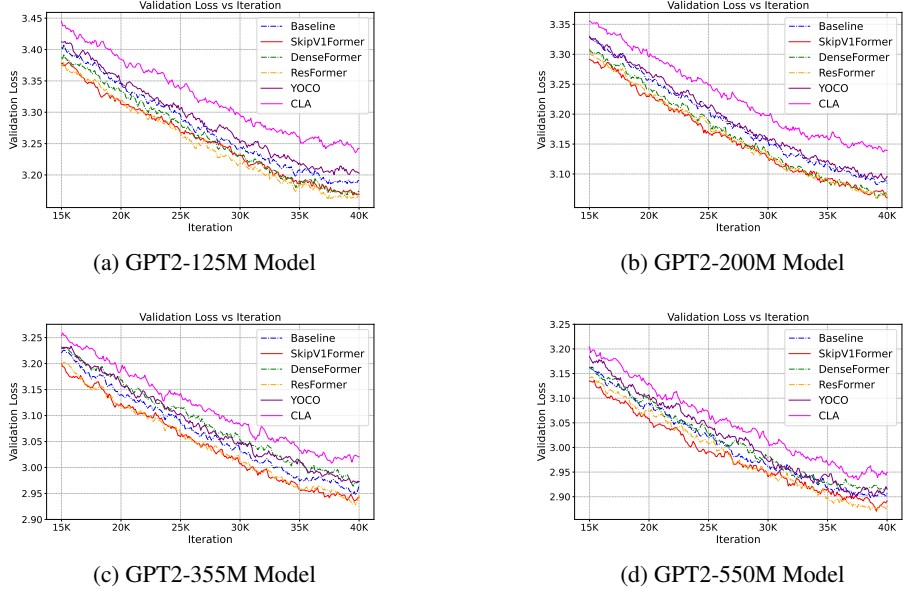

| (a) GPT2-125M Model | (b) GPT2-200M Model |
|---|---|
| (c) GPT2-355M Model | (d) GPT2-550M Model |

Figure 4: Validation loss curves across four GPT-2 model sizes (125M, 200M, 355M, 550M) for six architectures plotted over training iterations. Solid lines indicate KV-cache–efficient variants (YOCO-V, CLA-V, SkipV1Former), while dashed lines correspond to non-KV-cache-efficient models (MHA Transformer, DenseFormer, ResFormer).

# 6 Experiments

In this section, we evaluate SkipV1Former on GPT-2 [3] and LLaMA models [4], examining primarily pre-training dynamics and losses, inference memory, uptraining, and integration with KV-efficient techniques.

## 6.1 Experiments on GPT-2 Series

**Setup.**    We pretrain the models on OpenWebText2 [51], an open-source replication of OpenWeb-TextCorpus comprising around 17B tokens. We also evaluate the pretrained checkpoints on a set of standard downstream tasks in a zero-shot manner. We experiment on four model scales: 1) GPT2-S (125 M) and GPT2-M (355 M), following the configurations in Radford et al. [2] 2) Two additional intermediate variants for finer granularity: 200M and 550M.

For the architectures, we choose MHA Transformer as the baseline model. We pick two recent feature-augmentation methods: DenseFormer [7] and ResFormer [25], and two KV-sharing methods: YOCO [15] and Cross-Layer Attention (CLA) [16] for comparisons. To ensure fair comparison in the KV-cache-saving regime, we consider V-only YOCO and V-only CLA, aligning with SkipV1Former's design. All models are trained using AdamW. More details are provided in Appendix B.1.

**Results.**    The per-iteration validation losses for six architectures during pretraining are presented in Figure 4. SkipV1Former matches ResFormer in terms of final loss—achieving the lowest value overall. Importantly, SkipV1Former consistently outperforms both the baseline model and DenseFormer, highlighting the benefit of our skip connection with the first-layer Value. In the same KV-saving regime, SkipV1Former significantly surpasses the lightweight cache-efficient methods YOCO and CLA. Additionally, the validation-loss curve for SkipV1Former is as smooth as those of the regular models, demonstrating that our skip connection does not introduce any additional training instability.

We further evaluate these pretrained models in a zero-shot setting across standard downstream tasks in Appendix C. SkipV1Former and ResFormer yield the strongest average performance.

| (a) Validation Loss and Perplexity | | | | | | (b) Val. Loss vs. Model Size |
|---|---|---|---|---|---|---|

| **Validation Loss** | | | | | |
|---|---|---|---|---|---|
| **Model Size** | 60M | 130M | 350M | 1B | 3B |
| Baseline | 3.549 | 3.229 | 2.931 | 2.723 | 2.664 |
| SkipV1Former | 3.504 | 3.192 | 2.885 | 2.711 | 2.652 |
| **Perplexity** | | | | | |
| **Model Size** | 60M | 130M | 350M | 1B | 3B |
| Baseline | 34.78 | 25.25 | 18.75 | 15.23 | 14.35 |
| SkipV1Former | 33.25 | 24.34 | 17.90 | 15.04 | 14.18 |

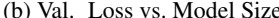
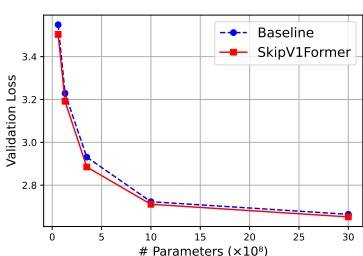

Figure 5: Validation Loss, Perplexity, and Loss Curve for LLaMA Models scaling from 60M to 3B. SkipV1Former consistently outperformed MHA Transformer with with reduced KV cache and fewer parameters.

## 6.2 Experiments on LLaMA Series

**Setup.** We further compare SkipV1Former and the baseline model MHA Transformer on LLaMA series [4] to validate the effectiveness of our architecture. We pre-train on C4 dataset, a colossal, cleaned version of Common Crawl's web crawl corpus [52]. We evaluate sizes from 60 M up to 3 B parameters to test scalability. More details are provided in Appendix B.2.

**Results.** Table 5a and Figure 5b report validation loss and perplexity across all sizes. SkipV1Former outperforms the baseline at every scale. On the small and medium models (60M–350M), it achieves notable reduction in validation loss of approximately 0.03–0.05. For larger models (1B–3B), SkipV1Former maintains a noticeable improvement regarding the model scale. Importantly, these performance gains come alongside a reduction of approximately 25% in KV cache usage and around 4% in total parameter count. Furthermore, Table 1 reports the downstream task accuracy of the 3B models, with results for other scales provided in Appendix C.3. SkipV1Former also surpasses the baseline in average downstream performance.

Table 1: Test accuracies (%) of 3B-scale models on downstream tasks, with overall average.

| Model | ARC-C | ARC-E | BoolQ | Hella | OBQA | PIQA | RTE | SciQ | Winogr | Avg |
|---|---|---|---|---|---|---|---|---|---|---|
| Baseline | 21.0 | **49.5** | 55.5 | 34.5 | **18.6** | 68.4 | 53.1 | 77.5 | 49.3 | 47.5 |
| SkipV1Former | 21.0 | 49.4 | **60.3** | **35.1** | 17.6 | **69.3** | **53.8** | **79.1** | **49.5** | **48.3** |

## 6.3 Inference

**GPU Memory.** To validate real-world savings, we measure and compare GPU memory consumption of SkipV1Former and a baseline MHA Transformer across different dimensions. All experiments are conducted on LLaMA variants using a single NVIDIA RTX A6000 GPU with 48 GB of memory, and the results are summarized in Figure 6.

Figure 6a plots KV cache size (in MB) as a function of sequence length. While both SkipV1Former and MHA Transformer exhibit linear cache growth, SkipV1Former's slope is ∼25 % lower than that of MHA across all LLaMA variants, aligning with our theoretical predictions. Figure 6b plots per-token KV cache (KB) against model size (ranging from 350M to 7B parameters). Again, SkipV1Former maintains a ∼25 % reduction at all scales, demonstrating uniform savings regardless of model capacity. Finally, Figure 6c provides a stacked breakdown of peak allocated memory in inference without activation offloading. SkipV1Former not only shrinks KV cache size but also lowers the overall peak memory use by a noticeable margin of ∼ 5.6GB.

**Prefill Time and Throughput.** We also evaluate inference speed in terms of prefill latency (time to build the KV cache) and decoding throughput (tokens/s with cached KV states). Figure 7 reports results across 60M–7B models. SkipV1Former reduces prefill time by 5–8 ms/100 tokens due to halved $W_V$ projections, and achieves throughput within ±2% of the baseline at practical batch sizes ($B = 8, 16$), with occasional gains. These modest but consistent improvements are achieved without

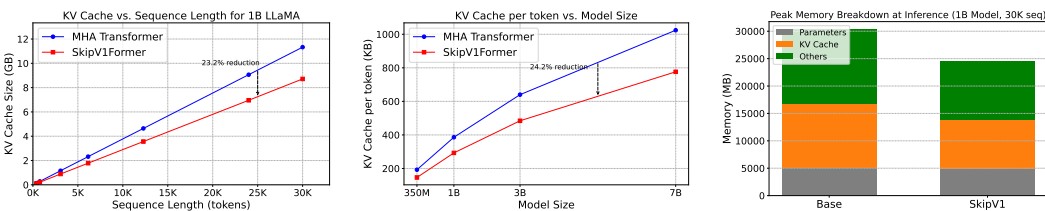

(a) KV cache size across different sequence length.

(b) Per-token KV cache as a function of model size.

(c) Memory breakdown at inference for 1B LLaMA.

Figure 6: Inference memory evaluation of SkipV1Former vs. standard MHA Transformer on LLaMA.

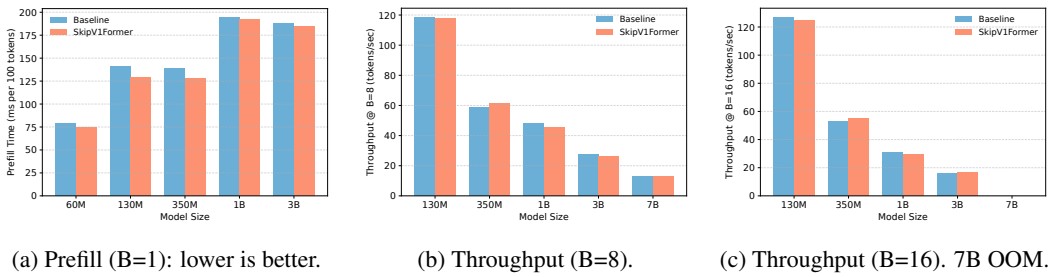

(a) Prefill (B=1): lower is better.

(b) Throughput (B=8).

(c) Throughput (B=16). 7B OOM.

Figure 7: Inference speed of SkipV1Former vs. baseline.

the additional computational overhead typically introduced by alternative KV-cache compression methods (e.g., SVD-based), underscoring SkipV1Former's efficiency. This overhead is expected to shrink with dedicated fused kernels. More details are provided in Appendix B.2.

### 6.4 Extensions

**Uptraining.** We uptrain SkipV1Formers from the 125M to 355M GPT2-MHA Transformer checkpoints using our uptraining strategy in Section 5.1. Results are shown in Figure 8, and further experimental details are included in Appendix B.1.

As shown in Figure 8, uptraining requires only 10%–15% of the original compute budget to reach the same validation loss as training from scratch. Furthermore, to match the final performance of a fully trained MHA Transformer, SkipV1Former requires only 5 %–10 % of the original training compute. The checkpoint conversion process incurs negligible resources compared to full training. We also compare alternative checkpoint conversion approaches in Appendix B.2.

**Combining with Existing Methods.** The skip connection mechanism of our SkipV1Former can be combined with other layer-replacement or low-rank methods such as GQA and MLA. We evaluate the SkipV1Former-GQA and SkipV1Former-MLA against their vanilla counterparts on the LlaMA 350M architecture on OpenWebText2. As shown in Table 2, the SkipV1Former-version of GQA and

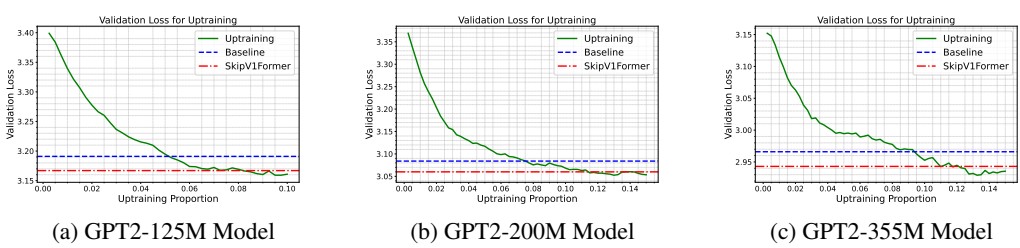

(a) GPT2-125M Model

(b) GPT2-200M Model

(c) GPT2-355M Model

Figure 8: Validation loss versus uptraining proportion for GPT2-125M, GPT2-200M, and GPT2-355M models. The horizontal dashed lines mark the final validation losses of the pretrained baseline (blue) and SkipV1Former (red) when trained from scratch.

Table 2: Performance and KV cache reduction when composing SkipV1Former with GQA and MLA on GPT2-355M. "Δ KV %" indicates cache reduction relative to the non-SkipV1 methods.

| Models | Val. Loss | PPL | Params | KV Bytes / token | Δ KV % |
|---|---|---|---|---|---|
| Transformer-GQA | 2.912 | 18.39 | 334.7 M | 98 304 B | — |
| SkipV1Former-GQA | 2.893 | 18.04 | 328.9 M | 74 752 B | −24.0% |
| Transformer-MLA | 2.896 | 18.10 | 337.4 M | 13 824 B | — |
| SkipV1Former-MLA | 2.888 | 17.95 | 334.4 M | 7 680 B | −44.4% |

MLA achieve lower validation loss and reduce the KV bytes per token by a large margin. The total parameter count is also reduced, benefiting from the lightweight nature of our skip mechanism. To further align with recent practice, we also experiment with GQA using a group-of-4 configuration on TinyLLaMA models. Results in Appendix B.2 show that SkipV1Former continues to outperform the baseline under this setting, confirming the robustness of the method across different GQA schemes. Additional results on combining SkipV1Former with YOCO are provided in Appendix C.5.

## 6.5 Ablations

**Skip-Head Ratio.** We vary the fraction of attention heads whose Value are replaced by first-layer Values—from 25 % up to 75 %—on GPT2-355M. As shown in Figure 9a, skipping more than 50 % of heads begins to hurt validation loss, while skipping fewer than 50 % also underperforms the 50% setting while consuming more memory. Hence, skipping exactly half of the heads seems to achieve an optimal tradeoff between model quality and memory saving. Additional experiments on LlaMA-1B models and discussions are provided in Appendix B.2.

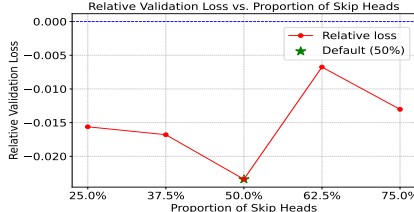

(a) Relative validation loss for cross-layer head ratio compared with baseline. The asterisk denotes the default ratio of skip heads.

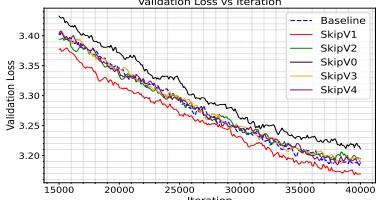

(b) Validation loss per iteration for cross-layer V at different layers and comparison with no skip (SkipV0) and baseline.

Figure 9: Comparison of validation loss under different head and layer configurations.

**Skip Connection with Different Layers.** We further investigate the performance of reusing Values of other layers (layer 2, 3, 4) as well as reusing no Values (SkipV0). As shown in Figure 9b, reusing second-layer Values (SkipV2) brings only marginal gains, and removing skip connections while halving the Value projection (SkipV0) severely degrades performance. Extending the skip to deeper layers (SkipV3 & SkipV4) similarly yields weaker results than SkipV1. Together, these findings highlight the unique importance of first-layer Values for cross-layer reuse.

## 7 Conclusions and Future Works

SkipV1Former demonstrates that strategic reuse of first-layer Value can simultaneously improve the model's representation capability and reduce KV cache size. This design enables deeper layers to access information typically lost to aggressive compression, thereby enhancing the model's inherent mesa-optimization. Across different model scales, SkipV1Former consistently outperforms MHA Transformer and related variants while using less KV memory. Its "uptraining" methodology and seamless integration with existing techniques further demonstrate its practical value and flexibility.

Looking forward, promising directions include kernel-level optimizations for faster inference, training-free transformations to retrofit existing models, and applications in fine-tuning. We leave these to future investigation and hope our study motivates broader exploration of cross-layer skip connections in large-scale Transformer architectures.

## Acknowledgements

K. Yuan is supported by the NSF China under Grants 92370121, 12301392, and W2441021. Z. Lin is supported by the NSF China under Grants 62276004 and the State Key Laboratory of General Artificial Intelligence.

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

# A Theoretical Analysis

## A.1 Results and Proofs

In this section, we give a formal statement of Theorem 1 and the detailed proof. We first restate the problem settings and assumptions.

We consider a set of in-context examples $\{(\mathbf{x}_i, \mathbf{y}_i)\}_i$, where $\mathbf{x}_i, \mathbf{y}_i \in \mathbb{R}^d$ and $\mathbf{y}_i = W^* \mathbf{x}_i + \varepsilon_i$. Here, $W^*$ is a shared coefficient matrix and $\varepsilon_i$ is the additive Gaussian noise for each sequence. The training input to the model is given by

$$E = (\mathbf{x}_1, \mathbf{y}_1, \cdots, \mathbf{y}_n, \mathbf{x}_{n+1}) \in \mathbb{R}^{d \times (2n+1)},$$

where each $\mathbf{x}_i$ is sampled independently from $\mathcal{N}(0, I_d)$, $W^* \sim \mathcal{N}(W_0, I_{d^2})$ for some deterministic $W_0 \in \mathbb{R}^{d \times d}$, and $\varepsilon_i \sim \mathcal{N}(0, \sigma^2 I_d)$ are i.i.d. Gaussian noise. For the model, we follow the common simplifications [41, 48, 42] considering a two-layer linear attention Transformer with two heads per layer and no residual connections. This simplified model preserves the key component—multi-head attention—and is sufficient to highlight the differences in information flow between the standard MHA Transformer and SkipV1Former. We train the model using only the in-context tokens $(\mathbf{x}_1, \mathbf{y}_1, \cdots, \mathbf{y}_n)$.

For the MHA Transformer, the output of each transformer block is

$$\mathrm{TF}_l(E) = \sum_{h=1}^{2} W_V^{l,h} E P \left( E^\top (W_K^{l,h})^\top W_Q^{l,h} E \right), \quad l = 1, 2, \tag{2}$$

where $P = \begin{pmatrix} I_{2n} & \mathbf{0} \\ \mathbf{0} & 0 \end{pmatrix} \in \mathbb{R}^{(2n+1) \times (2n+1)}$, $W_Q^{l,h}, W_K^{l,h}, W_V^{l,h} \in \mathbb{R}^{d \times d}$ for $l, h = 1, 2$, and we omit $W_O$ since it can be incorporated into $W_V^{l,h}$, making no difference in expressivity analysis.

Denote by $\tau$ the distribution over input sequences $E$. For SkipV1Former with 1 skip head, the output can be expressed as:

$$
\begin{aligned}
E^{\mathrm{out}} = {} & W_V^{2,1} \mathrm{TF}_1(E) P \left( \mathrm{TF}_1(E)(W_K^{2,1})^\top W_Q^{2,2} \mathrm{TF}_1(E) \right) \\
& + W_V^{2,2} E P \left( \mathrm{TF}_1(E)(W_K^{2,2})^\top W_Q^{2,2} \mathrm{TF}_1(E) \right),
\end{aligned}
\tag{3}
$$

where $\mathrm{TF}_1(\cdot)$ is defined as in Eq. (2). The training loss function is:

$$L_k \left( \{W_Q^{l,h}, W_K^{l,h}, W_V^{l,h}\}, H \right) = \mathbb{E}_\tau \left\| H E_{:,-1}^{\mathrm{out}} - \mathbf{y}_{n+1} \right\|_F^2, \tag{4}$$

where $k = 1$ refers to the loss of standard Transformer, and $k = 2$ refers to that of SkipV1Former.

As stated in Section 4, we admit the copying mechanism and introduce the following assumption:

**Assumption 1.** *The output of the first attention block is*

$$\mathrm{TF}_1(E) = \left( \mathbf{x}_1, \begin{pmatrix} A\mathbf{x}_1 \\ B\mathbf{y}_1 \end{pmatrix}, \cdots, \mathbf{x}_n, \begin{pmatrix} A\mathbf{x}_n \\ B\mathbf{y}_n \end{pmatrix}, \begin{pmatrix} A\mathbf{x}_{n+1} \\ \mathbf{0} \end{pmatrix} \right),$$

*where $A \in \mathbb{R}^{a \times d}, B \in \mathbb{R}^{(d-a) \times d}$ and $0 \leq a \leq d$.*

Assumption 1 is close to real scenarios, as the embedding dimension of attention is kept the same as that of token embedding [1]. Although we do not consider causal mask explicitly, we assume that the copying mechanism is conducted on only $\mathbf{y}_i$s with previous tokens, maintaining the causal structure. In Assumption 1, the information of the raw samples are compressed as $A\mathbf{x}$ and $B\mathbf{y}$. Intuitively, MHA can only solve the compressed regression problem with sample-label pairs $(A\mathbf{x}, B\mathbf{y})$, whereas SkipV1Former can access the uncompressed $\mathbf{x}$ and $\mathbf{y}$ and solve the original problem and attain lower loss.

Under Assumption 1, we drop the superscript $l$ in Eq. (2) and (3). The formal version of our main theorem is stated as follows:

**Theorem 2** (Formal). *Assume that $\sigma_d(W_0) > 2\sqrt{d}\sigma$ and $d$ is sufficiently large. Under Assumption 1, there exists a constant $c > 0$ independent of $A, B, W_Q^h, W_K^h, W_V^h, H$, such that*

$$\min_{W_Q^h, W_K^h, W_V^h, H} L_2(\{W_Q^h, W_K^h, W_V^h\}, H) \leq \min_{W_Q^h, W_K^h, W_V^h, H} L_1(\{W_Q^h, W_K^h, W_V^h\}, H) - c,$$

*where $L_1$ and $L_2$ are defined as in Eq. (4).*

The proof follows the analysis framework in [48]. We begin by rewriting the loss function in a more convenient form. Define $W^h = HW_V^h, M^h = (W_K^h)^\top W_Q^h \begin{pmatrix} A \\ \mathbf{0} \end{pmatrix}$ and set

$$G_{1,D} = \sum_{i=1}^n \mathbf{x}_i \mathbf{x}_i^\top + \begin{pmatrix} A\mathbf{x}_i \mathbf{x}_i^\top A^\top & A\mathbf{x}_i \mathbf{y}_i^\top B^\top \\ B\mathbf{y}_i \mathbf{x}_i^\top A^\top & B\mathbf{y}_i \mathbf{y}_i^\top B^\top \end{pmatrix}$$

for each input sequence $D = (\mathbf{x}_1, \mathbf{y}_1, \cdots, \mathbf{y}_n)$. Then the loss of MHA Transformer can be simplified as

$$L_1(A, B, W, M) = \mathbb{E}_\tau \left\| \sum_{h=1}^2 W^h G_{1,D} M^h \mathbf{x}_{n+1} - \mathbf{y}_{n+1} \right\|_F^2. \tag{5}$$

Similarly, loss of SkipV1Former can be expressed as

$$L_2(A, B, W, M) = \mathbb{E}_\tau \left\| \sum_{h=1}^2 W^h G_{2,D}^h M^h \mathbf{x}_{n+1} - \mathbf{y}_{n+1} \right\|_F^2, \tag{6}$$

where $G_{2,D}^1 = G_{1,D}$ and $G_{2,D}^2 = \sum_{i=1}^n \mathbf{x}_i \mathbf{x}_i^\top + \mathbf{y}_i \begin{pmatrix} A\mathbf{x}_i \\ B\mathbf{y}_i \end{pmatrix}^\top$.

To prove the theorem, we present the following lemmas.

**Lemma 1.** *Let $\hat{W}_D$ be the solution to ridge regression with regularization strength $\sigma^2$ on the exemplars $(\mathbf{x}_1, \cdots, \mathbf{y}_n, \mathbf{x}_{n+1})$. For any matrices $A$ and $B$, there exists a constant $C > 0$ independent of $A, B, W, M$, such that*

$$L_k(A, B, W, M) = C + \mathbb{E}_D \left\| \sum_{h=1}^2 W^h G_{i,D} M^h - \hat{W}_D \right\|_F^2, \quad k = 1, 2.$$

*Proof.* We denote

$$W = (W^1, W^2), \quad \tilde{G}_{1,D} = \begin{pmatrix} G_{1,D} & \\ & G_{1,D} \end{pmatrix}, \quad \tilde{G}_{2,D} = \begin{pmatrix} G_{2,D}^1 & \\ & G_{2,D}^2 \end{pmatrix}, \quad M = \begin{pmatrix} M^1 \\ M^2 \end{pmatrix}.$$

Using the law of total expectation, the loss can be written as

$$L_k(A, B, W, M) = \mathbb{E}_{D, \mathbf{x}_{n+1}} \mathbb{E}_{\mathbf{y}_{n+1}} \left[ \|\mathbf{y}_{n+1} - W\tilde{G}_{k,D} M\mathbf{x}_{n+1}\|_F^2 \mid D, \mathbf{x}_{n+1} \right], \quad k = 1, 2.$$

Define the function

$$g(U) = \mathbb{E}_{\mathbf{y}_{n+1}} \left[ \|U\mathbf{x}_{n+1} - \mathbf{y}_{n+1}\|_F^2 \mid D, \mathbf{x}_{n+1} \right].$$

Since $g(U)$ is a convex function, the minimizer of the function exists and we denote (any of) it by $\hat{W}_D$. The first-order optimality condition gives

$$\mathbf{0} = \nabla_U g(U) = \mathbb{E}_{\mathbf{y}_{n+1}} \left[ 2(U\mathbf{x}_{n+1} - \mathbf{y}_{n+1})\mathbf{x}_{n+1}^\top \mid D, \mathbf{x}_{n+1} \right].$$

For any matrix $U$, by taking the dot product of both sides with $U - \hat{W}_D$, we obtain

$$\mathbb{E}_{\mathbf{y}_{n+1}} \left[ 2(U\mathbf{x}_{n+1} - \mathbf{y}_{n+1})\mathbf{x}_{n+1}^\top (U - \hat{W}_D)^\top \mid D, \mathbf{x}_{n+1} \right] = \mathbf{0}. \tag{7}$$

Therefore, we have the following decomposition

$$\mathbb{E}_{\mathbf{y}_{n+1}}\left[\|\mathbf{y}_{n+1} - W\tilde{G}_{k,D}M\mathbf{x}_{n+1}\|_F^2 \,\Big|\, D, \mathbf{x}_{n+1}\right]$$

$$= \mathbb{E}_{\mathbf{y}_{n+1}}\left[\|\mathbf{y}_{n+1} - \hat{W}_D\mathbf{x}_{n+1} + \hat{W}_D\mathbf{x}_{n+1} - W\tilde{G}_{k,D}M\mathbf{x}_{n+1}\|_F^2 \,\Big|\, D, \mathbf{x}_{n+1}\right]$$

$$= \mathbb{E}_{\mathbf{y}_{n+1}}\left[\|\mathbf{y}_{n+1} - \hat{W}_D\mathbf{x}_{n+1}\|_F^2 + \|\hat{W}_D\mathbf{x}_{n+1} - W\tilde{G}_{k,D}M\mathbf{x}_{n+1}\|_F^2 \,\Big|\, D, \mathbf{x}_{n+1}\right]$$

$$+ 2\mathbb{E}_{\mathbf{y}_{n+1}}\left[(\mathbf{y}_{n+1} - \hat{W}_D\mathbf{x}_{n+1})(\hat{W}_D\mathbf{x}_{n+1} - W\tilde{G}_{k,D}M\mathbf{x}_{n+1})^\top \,\Big|\, D, \mathbf{x}_{n+1}\right].$$

By setting $U = W\tilde{G}_{k,D}M$ in Eq. (7), the last term vanishes. Hence,

$$L_k(A, B, W, M) = \mathbb{E}_{D,\mathbf{x}_{n+1}}\left[\mathbb{E}_{\mathbf{y}_{n+1}}\left[\|\hat{W}_D\mathbf{x}_{n+1} - W\tilde{G}_{k,D}M\mathbf{x}_{n+1}\|_F^2 \,\Big|\, D, \mathbf{x}_{n+1}\right]\right] + C,$$

where

$$C = \mathbb{E}_{D,\mathbf{x}_{n+1}}\mathbb{E}_{\mathbf{y}_{n+1}}\left[\|\mathbf{y}_{n+1} - \hat{W}_D\mathbf{x}_{n+1}\|_F^2 \,\Big|\, D, \mathbf{x}_{n+1}\right]$$

is independent of $A, B, W, M$.

Finally, since $\mathbf{x}_{n+1} \sim \mathcal{N}(0, I_d)$, we have

$$L_k(A, B, W, M) = C + \mathbb{E}_D\left\|\hat{W}_D - W\tilde{G}_{k,D}M\right\|_F^2.$$

$\square$

We denote $W^h = (W_1^h, W_2^h)$ and $M^h = (M_1^h, M_2^h)$, where $W_1^h, M_1^h \in \mathbb{R}^{d\times a}$ and $W_2^h, M_2^h \in \mathbb{R}^{d\times(d-a)}$. With these notations, the following lemma provides a further simplification of the loss for both the standard MHA Transformer and the SkipV1Former.

**Lemma 2.** *For the standard Transformer, there exists a constant $C_1(A, B, W, M) > 0$ such that*

$$L_1(A, B, W, M) = C + C_1 + \mathbb{E}_D\left\|H_1 - \hat{W}_D\right\|_F^2,$$

*where*

$$H_1 = \sum_{h=1}^2\left(W_1^h AXY^\top B^\top M_2^h + W_2^h BYX^\top A^\top M_1^h\right).$$

*Similarly, for the SkipV1Former, there exists a constant $C_2(A, B, W, M) > 0$ such that*

$$L_2(A, B, W, M) = C + C_2 + \mathbb{E}_D\left\|H_2 - \hat{W}_D\right\|_F^2,$$

*where*

$$H_2 = \sum_{h=1}^2 W^h YX^\top A^\top M_1^h.$$

*Proof.* From Lemma 1 and the definition of $\tilde{G}_{1,D}$ ( $G_{1,D}$), we have

$$L_1(A, B, W, M) = C + \mathbb{E}_D\left\|\hat{W}_D - W\tilde{G}_{1,D}M\right\|_F^2$$

$$= C + \mathbb{E}_D\left\|\hat{W}_D - \sum_{h=1}^2 W^h\left(XX^\top + \begin{pmatrix}AX\\BY\end{pmatrix}\begin{pmatrix}X^\top A^\top & Y^\top B^\top\end{pmatrix}\right)M^h\right\|_F^2. \tag{8}$$

For any fixed $A, B, W, M$, define

$$N_1(X) = \sum_{h=1}^2\left(W^h XX^\top M^h + W_1^h AXX^\top A^\top M_1^h\right),$$

$$N_2(Y) = \sum_{h=1}^2 W_2^h BYY^\top B^\top M_2^h, \tag{9}$$

$$N_3(X, Y) = \sum_{h=1}^2\left(W_1^h AXY^\top B^\top M_2^h + W_2^h BYX^\top A^\top M_1^h\right).$$

Then Eq. (8) becomes

$$L_1(A, B, W, M) = C + \mathbb{E}_D \left\| \hat{W}_D - N_1(X) - N_2(Y) - N_3(X, Y) \right\|_F^2.$$

From the definition of $\hat{W}_D$, it is known that $\hat{W}_D = YX^\top(XX^\top + \sigma I_d)^{-1}$ almost everywhere. Using this identity, we observe that $\hat{W}_D - N_3(X, Y)$ is an odd-degree function in $Y$, while $N_1(X) + N_2(Y)$ is an even-degree function in $Y$. Hence,

$$L_1(A, B, W, M) = C + \mathbb{E}_D \left\| \hat{W}_D - N_3(X, Y) \right\|_F^2 + \mathbb{E}_D \left\| N_1(X) + N_2(Y) \right\|_F^2.$$

The first part of the lemma follows by setting $H_1 = N_3$ and $C_1 = \mathbb{E}_D \left\| N_1(X) + N_2(Y) \right\|_F^2$.

Similarly, for the SkipV1Former, we can decompose $W\tilde{G}_{2,D}M$ into components of odd and even degree in $Y$. A direct calculation yields

$$L_2(A, B, W, M) = C + \mathbb{E}_D \left\| \hat{W}_D - N_3'(X, Y) \right\|_F^2 + \mathbb{E}_D \left\| N_1'(X) + N_2'(Y) \right\|_F^2,$$

where

$$
\begin{aligned}
N_1'(X) &= \sum_{h=1}^{2} \left( W^h XX^\top M^h + W_1^1 AXX^\top A^\top M_1^1 \right), \\
N_2'(Y) &= W_2^1 BYY^\top B^\top M_2^1 + W^2 YY^\top B^\top M_2^2, \\
N_3'(X, Y) &= W_1^1 AXY^\top B^\top M_2^1 + W_2^1 BYX^\top A^\top M_1^1 + W^2 YX^\top A^\top M_1^2.
\end{aligned}
\tag{10}
$$

The second part of the lemma follows by taking $H_2 = N_3'$ and $C_2 = \mathbb{E}_D \left\| N_1'(X) + N_2'(Y) \right\|_F^2$.

$\square$

Next, for any matrices $\tilde{W}^h = (\tilde{W}_1^h, \tilde{W}_2^h) \in \mathbb{R}^{d \times 2d}$ and $\tilde{M}^h = \begin{pmatrix} \tilde{M}_1^h \\ \tilde{M}_2^h \end{pmatrix} \in \mathbb{R}^{2d \times d}$ for $h = 1, 2$, we define

$$h(\tilde{W}, \tilde{M}) = \sum_{h=1}^{2} \tilde{W}^h \begin{pmatrix} & XY^\top \\ YX^\top & \end{pmatrix} \tilde{M}^h.$$

We focus on

$$\tilde{L}(\tilde{W}, \tilde{M}) = \mathbb{E}_D \left\| \hat{W}_D - h(\tilde{W}, \tilde{M}) \right\|_F^2 \tag{11}$$

in the following analysis. We first present Lemmas 3 and 4 from [48] and treat them as one lemma as follows.

**Lemma 3** (Lemma 3 and 4 in [48]). *There exists scalars $t_1$ and $t_2$, such that for any $i$, it holds that*

$$\mathbb{E}_D \left[ X(Y^T)_{:,i}(Y)_{i,:} X^T \right] = t_1 I_{d^2}, \quad \mathbb{E}_D \left[ X(Y^T)_{:,i}(\hat{W}_D)_{i,:} \right] = t_2 I_{d^2},$$

*where $Y_{i,:}$ denotes the $i$-th row and $Y_{:,i}$ denotes the $i$-th column of $Y$. Moreover, for any $i$, by setting*

$$\lambda_i = \frac{\mathbb{E}_D (\hat{W}_D)_{i,:} X(Y^T)_{:,i}}{\mathbb{E}_D (Y)_{i,:} X^T X(Y^T)_{:,i}},$$

*it holds that*

$$\mathbb{E}_D \left[ \lambda_i X(Y^T)_{:,i}(Y)_{i,:} X^T - X(Y^T)_{:,i}(\hat{W}_D)_{i,:} \right] = \mathbf{0}.$$

**Lemma 4.** *There exists a constant $C_3 > 0$ which is independent of $W, M$, such that*

$$\tilde{L}(\tilde{W}, \tilde{M}) = C_3 + \mathbb{E}_D \left\| \Lambda YX^T - h(\tilde{W}, \tilde{M}) \right\|_F^2,$$

*where $\Lambda = \mathrm{diag}\{\lambda_1, \cdots, \lambda_d\}$ and*

$$\lambda_i = \frac{\mathbb{E}_D (\hat{W}_D)_{i,:} X(Y^T)_{:,i}}{\mathbb{E}_D (Y)_{i,:} X^T X(Y^T)_{:,i}}.$$

*Proof.* We denote $\tilde{L}'(W, M) = \mathbb{E}_D \left\| \Lambda Y X^T - h(\tilde{W}, \tilde{M}) \right\|_F^2$ and $G'_D = \begin{pmatrix} & XY^T \\ YX^T & \end{pmatrix}$ for simplicity. Our goal is to show that

$$\nabla_{(\tilde{W}, \tilde{M})} \tilde{L}(\tilde{W}, \tilde{M}) = \nabla_{(\tilde{W}, \tilde{M})} \tilde{L}'(\tilde{W}, \tilde{M}).$$

We first consider the gradient with respect to $\tilde{W}^1$:

$$\nabla_{\tilde{W}^1} \tilde{L}(\tilde{W}, \tilde{M}) = 2\mathbb{E}_D \left[ (h(\tilde{W}, \tilde{M}) - \hat{W}_D)(\tilde{M}^1)^\top G'_D \right],$$

$$\nabla_{\tilde{W}^1} \tilde{L}'(\tilde{W}, \tilde{M}) = 2\mathbb{E}_D \left[ (h(\tilde{W}, \tilde{M}) - \Lambda Y X^T)(\tilde{M}^1)^\top G'_D \right].$$

Thus it suffices to prove

$$\mathbb{E}_D \left[ \hat{W}_D (\tilde{M}^1)^\top G'_D \right] = \mathbb{E}_D \left[ \Lambda Y X^\top (\tilde{M}^1)^\top G'_D \right], \tag{12}$$

which breaks down into:

$$\mathbb{E}_D \left[ \hat{W}_D (\tilde{M}_1^1)^\top XY^\top \right] = \mathbb{E}_D \left[ \Lambda Y X^\top (\tilde{M}_1^1)^\top XY^\top \right], \tag{13}$$

$$\mathbb{E}_D \left[ \hat{W}_D (\tilde{M}_2^1)^\top YX^\top \right] = \mathbb{E}_D \left[ \Lambda Y X^\top (\tilde{M}_2^1)^\top YX^\top \right]. \tag{14}$$

We first prove Eq. (13). it suffices to show:

$$\text{tr} \left( \mathbb{E}_D \left[ (\tilde{M}_1^1)^\top (XY^\top)_{:,j} (\hat{W}_D)_{i,:} \right] \right) = \text{tr} \left( \mathbb{E}_D \left[ (\tilde{M}_1^1)^\top (XY^\top)_{:,j} (\Lambda Y X^\top)_{i,:} \right] \right), \quad \forall i, j.$$

When $i \neq j$, using the i.i.d. Gaussianity of $W$ and $\varepsilon_i$, we obtain:

$$\mathbb{E}_D \left[ (Y^\top)_{:,j} Y_{i,:} \mid X \right] = \mathbf{0}.$$

Thus we have

$$\mathbb{E}_D \left[ (XY^\top)_{:,j} (\hat{W}_D)_{i,:} \right] = \mathbb{E}_D \left[ (XY^\top)_{:,j} (\Lambda Y X^\top)_{i,:} \right], \quad \forall i \neq j.$$

When $i = j$, by Lemma 3, we also have

$$\mathbb{E}_D \left[ (XY^\top)_{:,i} (\hat{W}_D)_{i,:} \right] = \mathbb{E}_D \left[ (XY^\top)_{:,i} (\Lambda Y X^\top)_{i,:} \right], \quad \forall i.$$

Thus Eq. (13) holds.

For Eq. (14), it is equivalent to

$$\mathbb{E}_D \left[ (\hat{W}_D)_{i,:} (\tilde{M}_2^1)^\top YX^\top \right] = \mathbb{E}_D \left[ (\Lambda Y X^\top)_{i,:} (\tilde{M}_2^1)^\top YX^\top \right], \quad \forall i,$$

which can be further written as

$$\mathbb{E}_D \left[ (\hat{W}_D)_{i,:} \sum_{j=1}^d ((\tilde{M}_2^1)^\top)_{:,j} (YX^\top)_{j,:} \right] = \mathbb{E}_D \left[ (\Lambda Y X^\top)_{i,:} \sum_{j=1}^d ((\tilde{M}_2^1)^\top)_{:,j} (YX^\top)_{j,:} \right], \quad \forall i.$$

By the commutativity of the dot product, the above expressions can be rearranged as

$$\mathbb{E}_D \left[ \sum_{j=1}^d (\tilde{M}_2^1)_{j,:} (\hat{W}_D^\top)_{:,i} (YX^\top)_{j,:} \right] = \mathbb{E}_D \left[ \sum_{j=1}^d (\tilde{M}_2^1)_{j,:} (XY^\top \Lambda)_{:,i} (YX^\top)_{j,:} \right].$$

From the above discussion, we obtain

$$\mathbb{E}_D \left[ (\hat{W}_D^\top)_{:,i} (YX^\top)_{j,:} \right] = \mathbb{E}_D \left[ (XY^\top \Lambda)_{:,i} (YX^\top)_{j,:} \right], \quad \forall i, j,$$

which completes the proof of Eq. (14).

It remains to show that $\nabla_{\tilde{M}^1} \tilde{L}(\tilde{W}, \tilde{M}) = \nabla_{\tilde{M}^1} \tilde{L}'(\tilde{W}, \tilde{M})$. Similarly, this is equivalent to showing

$$\mathbb{E}_D \left[ (G'_D)^\top (\tilde{W}^1)^\top \hat{W}_D \right] = \mathbb{E}_D \left[ (G'_D)^\top (\tilde{W}^1)^\top \Lambda Y X^\top \right].$$

This can be shown using essentially the same technique as above. In particular, by setting $\tilde{W} = \mathbf{0}$, we have

$$C_3 = \mathbb{E}_D \left\| \hat{W}_D \right\|_F^2 - \mathbb{E}_D \left\| \Lambda Y X^\top \right\|_F^2.$$

$\square$

We now turn back to the proof of Theorem 1.

*Proof of Theorem 1.* By the definition of loss of standard MHA Transformer and SkipV1Former in Eq. (5) and (6) and the form of $\tilde{L}(\tilde{W}, \tilde{M})$ in Eq. (11), we immediately obtain

$$\min_{A,B,W,M} \mathbb{E}_D \left\| \hat{W}_D - N_3'(X,Y) \right\|_F^2 \geq \min_{\tilde{W}, \tilde{M}} \tilde{L}(\tilde{W}, \tilde{M}).$$

On the other hand, when

$$A = I, B = \mathbf{0}, W^1 = -\frac{1}{2}\Lambda, W^2 = \Lambda, M^1 = I, M^2 = I,$$

it follows from Eq. (10) that

$$N_1'(X) = \mathbf{0}, N_2'(Y) = \mathbf{0}, N_3'(X,Y) = \Lambda Y X^\top.$$

Therefore, from Lemma 4, we know that

$$\min_{A,B,W,M} L_2(A,B,W,M) = C + C_3.$$

For the MHA Transformer, when $a = d$, implying $B = \mathbf{0}$, we can conclude from Lemma 2 that

$$L_1(A,B,W,M) \geq C + \mathbb{E}_D \left\| \hat{W}_D \right\|_F^2 = C + C_3 + \mathbb{E}_D \left\| \Lambda Y X^\top \right\|_F^2.$$

When $a < d$, noting from the definition that $M^h = W_{QK}^h \begin{pmatrix} A \\ \mathbf{0} \end{pmatrix}$, the rank of $N_3(X,Y)$ is at most $a$. Moreover, by Lemma 4 and the fact that $N_3(X,Y)$ is a special case of $h(\tilde{W}, \tilde{M})$, it holds

$$\mathbb{E}_D \left\| \hat{W}_D - N_3(X,Y) \right\| = C_3 + \mathbb{E}_D \left\| \Lambda Y X^\top - N_3(X,Y) \right\|$$
$$\geq C_3 + \mathbb{E}_D \| \Lambda Y X^\top - R_a(\Lambda Y X^\top) \|_F^2,$$

where $R_a(\Lambda Y X^\top)$ denotes the best rank-$a$ approximation to $\Lambda Y X^\top$ under the Frobenius norm. Denote by $c = \mathbb{E}_D \| \Lambda Y X^\top - R_a(\Lambda Y X^\top) \|_F^2$, which is a constant independent of $A, B, W, M$. Since $C_2(A,B,W,M) > 0$, we have

$$L_2(A,B,W,M) \geq C + C_3 + c = L_2'(A,B,W,M) + c.$$

It remains to show that $c > 0$. First, we show that $|\lambda_i| > 0$. Recall that $\hat{W}_D = YX^\top(XX^\top + \sigma^2 I)^{-1}$, so we have

$$\mathbb{E}_D \left[ (\hat{W}_D)_{i,:} X(Y^\top)_{:,i} \right] = \mathbb{E}_D \left[ Y_{i,:} X^\top (XX^\top + \sigma^2 I)^{-1} X(Y^\top)_{:,i} \right],$$
$$= \mathbb{E}_D \left[ W_{i,:} XX^\top (XX^\top + \sigma^2 I)^{-1} XX^\top (W^\top)_{:,i} \right] + \mathbb{E}_D \left[ \varepsilon_{i,:} X^\top (XX^\top + \sigma^2 I)^{-1} X(\varepsilon^\top)_{:,i} \right].$$

By diagonalizing $XX^\top$, one can show that

$$XX^\top (XX^\top + \sigma^2 I)^{-1} XX^\top \succeq XX^\top - \sigma^2 I.$$

Thus, we can derive

$$\mathbb{E}_X \left[ XX^\top (XX^\top + \sigma^2 I)^{-1} XX^\top \right] \succeq \mathbb{E}_X [XX^\top - \sigma^2 I] = (d - \sigma^2) I.$$

We also have $X^\top (XX^\top + \sigma^2 I)^{-1} X \succeq \mathbf{0}$. Combining these results, we conclude that $|\lambda_i| > 0$. Now, since $\Lambda$ is a full-rank matrix, we obtain

$$\mathbb{E}_D \| \Lambda Y X^\top - R_a(\Lambda Y X^\top) \|_F^2 \geq \min_i |\lambda_i|^2 \cdot \mathbb{E}_D \| Y X^\top - R_a(Y X^\top) \|_F^2.$$

By the Eckart-Young theorem [53], we have

$$\mathbb{E}_D \| Y X^\top - R_a(Y X^\top) \|_F^2 \geq \mathbb{E}_D \left| \sum_{i=a+1}^d \sigma_i(Y X^\top) \right|^2 \geq \mathbb{E}_D \left[ \sigma_d^2(Y X^\top) \right],$$

where $\sigma_i(YX^\top)$ denotes the $i$-th singular value of $YX^\top$. Since $YX^\top = WXX^\top + \varepsilon X^\top$, by Weyl's inequality, we further have

$$\left| \sigma_d(YX^\top) - \sigma_d(W^* XX^\top) \right| \leq \|\varepsilon X\|_2.$$

For $\sigma_d(W^* XX^\top)$, since $\sigma_d(AB) \geq \sigma_d(A)\sigma_d(B)$, we have

$$\sigma_d(W^* XX^\top) \geq \sigma_d(W^*)\sigma_d^2(X).$$

By Edelman's Theorem [54], for $X \sim \mathcal{N}(\mathbf{0}, I_{d^2})$, it holds that $\sigma_d(X) \sim \Theta\left(\frac{1}{\sqrt{d}}\right)$ when $d$ is large. Thus, we have

$$\sigma_d(W^* XX^\top) \geq \sigma_d(W^*)\Omega\left(d^{-1}\right),$$

with high probability. Since $W \sim (U, \sigma I_{d^2})$, from the assumption that $\sigma_d(W_0) > 2\sqrt{d}\sigma$, we have

$$\mathbb{E}_D\left[\sigma_d(W^* XX^\top) - \|\varepsilon X\|_2\right] > 0.$$

Combining all the above results, we have shown that $c > 0$, thus completing the proof.

$\square$

## A.2  Discussions

In this section, we discuss the implications and relations to the analysis in other works of our theoretical findings. A high-level basis of our analysis centers on the interplay between Transformer architecture and the tasks it solves, in particular, auto-regressive tasks. Appendix C.6 preliminarily shows that SkipV1Former fails to boost performance on non-autoregressive tasks, indirectly support its design's focus on auto-regressive. In addition, the importance of the first layer is also confirmed in [55]. They similarly attribute in-context learning to a two-phase process: the first layer preprocesses context examples, while deeper layers perform iterative optimization steps.

In another related work, Zhou et al. [25] explain ResFormer's (akin to our skip connection with first-layer Values) gains via its ability in alleviating attention concentration in multi-head attention. While it is not clear whether there is a theoretical connection between their explanations and our analysis, Zhou et al. also confirm first-layer information loss, which is consistent with our analysis of copying-induced compression.

Due to the essential difficulty in analyzing the behavior of multi-layer Transformers in ICL [42], we focus on a simplified two-layer model without residual connection in Theorem 2-a tractable yet still instructive framework. While one may worry that omitting residual connections could affect deeper layers, several clues suggest that our analysis may still extend to standard Transformers. First, the "linear copying" mechanism in Assumption 1 appears at the first-layer output of standard deep Transformers *with* residual connections (see Figure 2 of [14]). Second, for simplified one-layer Transformers with and without residuals, it has been shown, respectively in [48] and [42], that both models implement a single gradient-descent step on a least-squares objective at optimality. In other words, removing residual connections does not appear to impair the representational capacity of Transformers for in-context learning, suggesting that our simplification is reasonable to an extent. Regarding the two-layer simplified model, if we assume that a multi-layer Transformer performs one gradient-descent step per layer, our analysis naturally generalizes to deeper models by applying the same proof technique as in Theorem 2. Furthermore, Appendix C.1 presents a linear-probe experiment on deeper models that provides empirical support for our analysis.

# B  Details and Additional Results for Section 6

## B.1  GPT-2 Series

**Pretraining.**   We provide implementation details of our GPT-2 pretraining setup, following the training framework in [7]. Table 3 presents the main hyperparameters of the model and optimizer used in training. The hidden dimension of the FFN is set as $4\times$ the embedding size. All the models are trained using BF16 format and AdamW optimizer [56] with $\beta_1 = 0.9$, $\beta_2 = 0.95$ and weight decay 0.1. We adopt a warmup ratio of 10% and a cosine annealing schedule with a decay factor of

| Params | Embed | Heads | Layers | Steps | Data Amount | Peak LR |
|--------|-------|-------|--------|-------|-------------|---------|
| 125M | 768 | 12 | 12 | 40K | 5B | 1e-3 |
| 200M | 1024 | 16 | 12 | 40K | 5B | 1e-3 |
| 355M | 1024 | 16 | 24 | 40K | 5B | 1e-3 |
| 550M | 1200 | 20 | 28 | 40K | 5B | 6e-4 |

Table 3: Hyperparameters of GPT-2 models in pretraining. "Data Amount" refers to the total number of tokens used during pretraining.

10% of the peak learning rate. The batchsize is 120 and the sequence length is 1024. We also use dropout as 0.2 and flash attention [57]. The experiments are conducted on 6×48 GB A6000 GPUs.

The architectures used in our experiments (DenseFormer, YOCO, CLA) are as follows:

- **DenseFormer**: DenseFormer enhances Transformer efficiency by introducing Depth Weighted Averaging (DWA) after each block. This mechanism computes a weighted average of the current and past representations, allowing the model to achieve lower perplexity without increasing its size.

- **YOCO (You Only Cache Once)**: YOCO proposes a decoder-decoder architecture that caches key-value pairs only once. It consists of a self-decoder and a cross-decoder, where the self-decoder computes the first $\frac{L}{2}$ layers using efficient attention and the cross-decoder computes the rest $\frac{L}{2}$ layers by replacing the $K, V$ in each of last $\frac{L}{2}$ with $\frac{L}{2}$'s $K, V$.

- **CLA (Cross-Layer Attention)**: CLA reduces the size of the key-value cache in Transformers by sharing Keys and Values for every $l$ adjacent layers. $l$ is typically set as 2 or 3.

To maintain the same KV-cache saving regime, we consider V-only YOCO, which does not incorporate efficient attention and only reuses the value in the last $\frac{L}{2}$ layers, and V-only CLA, which shares the value across every two adjacent layers, achieving a 25% reduction in KV cache size compared to the baseline.

**Uptraining.** For uptraining GPT2-125M to GPT2-355M models, we perform a grid search over learning rates {8e-5, 1e-4, 1.5e-4, 2e-4}, where 1e-4 is the final learning rate of our pretraining. Among these, a learning rate of 1.5e-4 yields the best performance consistently across all three model sizes. Dropout is disabled during uptraining, while all other hyperparameters remain consistent with the pretraining configuration. To initialize the uptrained models, we also experiment with several strategies for converting the pretrained checkpoints:

- **Mean VO:** Apply average pooling over every two head projections and their corresponding columns in $W_O$, and zero out the remaining columns.

- **Top V:** Select the $H'$ rows with the largest norm of head projections $W_V$ to be the new head projections.

- **Top VO:** Select the $H'$ head projections in $W_V$ and the $H'$ output columns in $W_O$ with the highest norms.

- **SVD:** Construct a best $H'$-rank approximation to the original $W_O W_V$ using SVD.

All the above approaches are conducted on every layer beyond the first. Empirically, all of the above strategies are outperformed by our method introduced in Section C.3, as evidenced by their higher initial losses. The comparisons of the converted checkpoints' initial loss are shown in Figure 10.

## B.2 LlaMA Series

**Pretraining.** We adopt the training framework from [58] and provide the following details regarding the models and hyperparameters. Table 4 summarizes the key hyperparameters of the models and optimizers used during training. All the models are trained using BF16 format and AdamW optimizer with $\beta_1 = 0.9$, $\beta_2 = 0.95$ and no dropout. For the 60M - 350M models, we apply no weight decay and gradient clipping, with a peak learning rate 2e-3. For the 1B and 3B models, we set weight decay to 0.1 and gradient clipping to 1.0, with a peak learning rate of 1e-3 and 5e-4, respectively. A

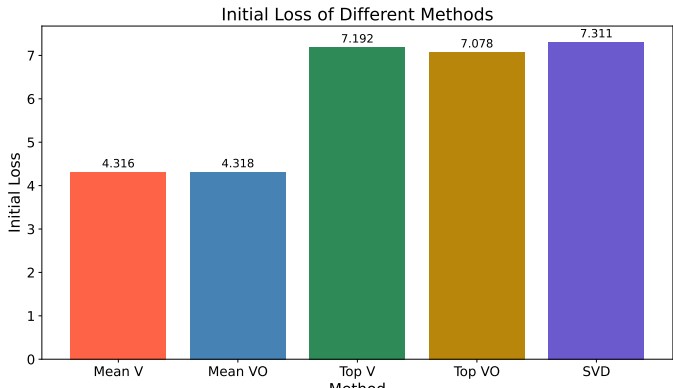

Figure 10: Comparison of initial loss values across different methods. The "Mean V" method corresponds to our baseline strategy described in the main text.

warmup ratio of 10% is used, along with a cosine annealing schedule decaying to 10% of the peak learning rate. The batchsize is 512 and sequence length is 1024 for all models. We also use Rotary Positional Embedding (RoPE) [59] and flash attention. The experiments are conducted on $8\times80$ GB A800 GPUs.

| Params | Hidden | Intermediate | Heads | Layers | Steps | Data Amount | Peak LR |
|---|---|---|---|---|---|---|---|
| 60M | 512 | 1376 | 8 | 8 | 2.5K | 1.3B | 2e-3 |
| 130M | 768 | 2048 | 12 | 12 | 5K | 2.6B | 2e-3 |
| 350M | 1024 | 2736 | 16 | 24 | 15K | 7.8B | 2e-3 |
| 1.3B | 2048 | 5461 | 32 | 24 | 25K | 13.1B | 1e-3 |
| 3B | 2560 | 6848 | 32 | 32 | 30K | 15.7B | 5e-4 |

Table 4: Hyperparameters of LlaMA models in pretraining. "Data Amount" refers to the total number of tokens used during pretraining.

We tune the learning rate from a set {2e-3, 1e-3, 5e-4} for all models and choose the best learning rate based on the validation loss of the baseline model. We observe that the influence of hyperparameters on SkipV1Former is nearly identical to that on the standard MHA Transformer.

To further assess robustness, we also pretrain LLaMA-1B models with different random seeds and perform a preliminary significance check. As shown in Table 5, SkipV1Former consistently outperforms the baseline by a similar margin under both seeds, indicating that the observed gains are not seed-dependent. In addition, similar trends are observed on TinyLLaMA-1.1B (see Table 6). These results confirm the improvement of SkipV1Former on accuracy.

**Combining with GQA and MLA.**   We first provide a brief introduction to GQA and MLA. GQA divides the query heads into $G$ groups. Each group shares a single KV pair, reducing memory usage while maintaining performance. Mathematically, for each group $g$, the attention output is computed as:

$$\text{head}_{i,g} = \text{Attention}(W_g^Q Q_i, W_g^K K, W_g^V V),$$

Table 5: Validation loss of LLaMA-1B models under different random seeds. Reported mean $\pm$ standard deviation is across three runs.

| Model | Seed 42 | Seed 100 | Seed 2025 | Mean $\pm$ Std |
|---|---|---|---|---|
| Baseline | 2.723 | 2.753 | 2.744 | $2.740 \pm 0.015$ |
| SkipV1Former | **2.711** | **2.743** | **2.732** | **$2.729 \pm 0.016$** |

Table 6: Validation loss for TinyLLaMA models with GQA-4 configuration.

| Model | 1.1B | 315M | 125M |
|---|---|---|---|
| TinyLLaMA (baseline) | 2.750 | 2.936 | 3.420 |
| TinyLLaMA-SkipV1 | **2.742** | **2.916** | **3.370** |

where $Q_i$ is the $i$-th query head, $W_g^Q, W_g^K, W_g^V$ are the projection matrices for the $g$-th group, $K$ and $V$ are the shared Key and Value for the group. The final output of GQA is computed in the same manner as that of MHA Transformer.

MLA introduces a low-rank compression of the Key and Value to reduce the memory footprint of the KV cache during inference. Instead of caching the full Keys and Values, MLA projects them into lower-dimensional latent spaces. Mathematically, let $X \in \mathbb{R}^{d \times n}$ be the input of an MLA layer. The inference of the layer can be expressed as

$$
\begin{aligned}
C^{KV} &= W^{DKV} X, \\
K^C &= W^{UK} C^{KV}, \\
K^R &= \text{RoPE}(W^{KR} X), \\
K &= \left[ K^R, K^C \right], \\
V^C &= W^{UV} C^{KV},
\end{aligned}
$$

where we omit the splitting of multi-heads. Here, $C^{KV} \in \mathbb{R}^{d_c \times n}$ is the compressed latent tensor for Keys and Values; $d_c \ll d$ indicates the KV compression dimension; $W^{DKV} \in \mathbb{R}^{d_c \times d}$ denotes the down-projection matrix; $W^{UK}, W^{UV} \in \mathbb{R}^{d \times d_c}$ are the up-projection matrices for the Keys and Values, respectively; $W^{KR} \in \mathbb{R}^{d_R \times d}$ is the matrix used to produce the decoupled key that carries Rotary Positional Embedding (RoPE); and $[\cdot, \cdot]$ dentoes concatenation. In MLA, only the $C^{KV}$ and $K^R$ need to be cached during generation.

We pretrain Base-GQA and Base-MLA, as well as their SkipV1Former counterparts, on LLaMA-350M with the OpenWebText2 dataset. The optimizer hyperparameters follow those used for GPT2-355M. In the experiments, GQA models share the same keys and values between every two heads for all layers, i.e., $G = H/2$. MLA models adopt $d_c = 256$ and $d_R = 32$.

For clarity, we detail how the total parameter counts in Table 2 are obtained. Each Transformer block contains an attention module and a feed-forward network (FFN). In the GPT2-355M setting, $d_{\text{model}} = 1024$, head dimension $= 64$, and FFN hidden size $d_{\text{ff}} = 4096$.

*GQA baseline.* With 24 layers and 16 query heads, we have $W_Q \in \mathbb{R}^{1024 \times 1024}$, $W_K, W_V \in \mathbb{R}^{1024 \times 512}$ (due to head sharing), and $W_O \in \mathbb{R}^{1024 \times 1024}$, giving $\sim$3.7M parameters per layer for attention. The FFN adds $\sim$8.4M per layer from $W_1 \in \mathbb{R}^{4096 \times 1024}$ and $W_2 \in \mathbb{R}^{1024 \times 4096}$. Over 24 layers this yields $\sim$101M (attention) + $\sim$201M (FFN), plus embeddings/norms for a total of 334.7M. SkipV1Former halves $W_V$ from layer 2 onward, i.e., $23 \times (1024 \times 256) \approx 6.0$M fewer parameters, giving 328.9M.

*MLA baseline.* Here Values use a low-rank latent $d_c = 256$, with $W_V^{\text{down}} \in \mathbb{R}^{1024 \times 256}$ and $W_V^{\text{up}} \in \mathbb{R}^{256 \times 1024}$ ($\sim$0.52M per layer), plus $W_Q, W_K, W_O$, so attention totals $\sim$3.9M per layer and FFN $\sim$8.4M. This gives about 93M + 201M = 337.4M overall. SkipV1Former-MLA halves $d_c$ to 128 from layer 2 onward, reducing $\sim$3.0M parameters, for a total of 334.4M.

As the above setting corresponds to the group-of-2 GQA scheme in LLaMA (v1), while more recent models such as LLaMA-2 and LLaMA-3 employ a group-of-4 configuration, where keys and values are further shared across four heads. To align with this more realistic setting, we additionally conduct experiments on TinyLLaMA models with group-of-4 GQA. Specifically, we compare SkipV1Former against the baseline on TinyLLaMA-1.1B, and also evaluate smaller 315M and 125M TinyLLaMA variants with the same modification strategy as used for LLaMA-130M/350M with LLaMA-1.1B.

The results, summarized in Table 6, show that SkipV1Former continues to outperform the baseline models by a margin comparable to that observed for GQA-2 (Table 2). This confirms that the effectiveness of cross-layer Value injection remains robust under more advanced GQA configurations.

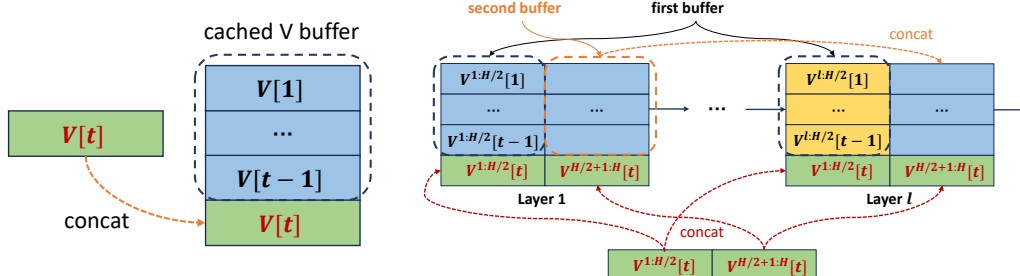

(a) Standard Transformer: KV cache extension during decoding. Each new token's Value is appended along the sequence dimension.

(b) SkipV1Former: KV caching with first-layer reuse. Deeper layers concatenate their current half-heads with the cached half-heads from the first layer.

Figure 11: Comparison of KV caching mechanisms during autoregressive decoding.

Table 7: Validation loss of LLaMA-1B models under different skip-head ratios.

| Skip Ratio | 25% | 50% (default) | 75% | Baseline |
|---|---|---|---|---|
| Val. Loss | 2.716 | **2.711** | 2.717 | 2.723 |

**KV Caching Mechanism in SkipV1Former.** During autoregressive decoding, SkipV1Former differs from the standard Transformer in how Value tensors are cached and retrieved.

*Standard Transformer.* For each new token $x_t \in \mathbb{R}^d$, the cached Value tensor is $V[1:t-1] \in \mathbb{R}^{bs \times H \times (t-1) \times d}$, which stores the Values of tokens $x_1, \ldots, x_{t-1}$. The Value of $x_t$ is simply appended along the sequence dimension, producing $V[1:t]$.

*SkipV1Former.* In contrast, SkipV1Former maintains two sets of cached buffers: (i) the first half of the heads for every layer, $V^{l,1:H/2}[1:t-1]$ for $l = 1, \ldots, L$, and (ii) the second-half heads of the first layer, $V^{1,H/2+1:H}[1:t-1]$. When processing $x_t$, these buffers are extended along the sequence dimension to include the new token, yielding $V^{l,1:H/2}[1:t]$ and $V^{1,H/2+1:H}[1:t]$. They are then concatenated along the head dimension to form the full set of Values used in attention at step $t$. An illustration is provided in Figure 11.

This design effectively reuses first-layer Value information across all deeper layers via per-head concatenation, which is absent in the standard Transformer. Conceptually, the difference is not in appending along the sequence dimension—which both models share—but in the additional cross-layer head stitching. While this introduces minor overhead compared to standard V-cache extension, it is an implementation detail rather than a kernel-level limitation, and can be optimized further with fused kernels.

**Ablations.** In the main text, we adopt a 50% skip-head ratio as the default setting. Since the total number of concatenated $V$ heads is fixed, a larger skip ratio implies a stronger reduction in the embedding dimension of $W_V$ from layer 2 onward. Skipping fewer than 50% of heads offers limited memory savings, while skipping all heads corresponds to SVFormer [25], which suffers from clear performance degradation. Intuitively, too few skipped heads restrict deeper layers by limiting the copying mechanism, whereas too many skipped heads hinder their ability to process the full data flow. Empirically, a 50% ratio achieves a near-optimal tradeoff between memory efficiency and model quality.

To further validate this finding, we extend the ablation to a larger 1B-parameter LLaMA model. Table 7 reports validation loss under 25%, 50%, and 75% skip ratios, together with the baseline. Consistent with the GPT2-355M results, the 50% ratio again provides a near-optimal tradeoff. Notably, the 75% ratio still improves upon the baseline while further reducing memory cost compared to the default 50% ratio, making it a viable option in highly memory-constrained scenarios.

# C  Additional Experiments

## C.1  Linear Probe

In this section, we validate our theoretical findings from Section 4 by examining the mesa-optimization behavior in Transformers using linear probes [60]. We follow the methodology outlined in [14, 43]. Specifically, we extract the output from each layer of a pretrained GPT2-125M model and train a linear probe on top. The probe consists of a layer normalization followed by a linear classification head. The learning rate is tuned over the set {1e-4, 2e-4, 3e-4}, and training is conducted for 4000 steps, with all other hyperparameters remaining consistent with those used during pretraining.

Figure 12 illustrates the validation loss of the probe across each layer. For both the baseline model and SkipV1Former, the probe loss decreases monotonically with layer depth, indicating progressively refined internal representations. The trend can be roughly divided into two phases:

- **Layers 0–6**: In these early layers, the probe loss fluctuates between the base and SkipV1Former models. This behavior suggests that the shallow layers are transforming the input into representations conducive to mesa-optimization [61].

- **Layers 7–11**: In these deeper layers, the probe loss for SkipV1Former shows a consistently steeper downward trend compared to the baseline. This indicates that SkipV1Former has accelerated the mesa-optimization process, resulting in more rapidly improving representations.

These observations empirically support our theoretical analysis, demonstrating that SkipV1Former enhances the efficiency of mesa-optimization in deeper layers.

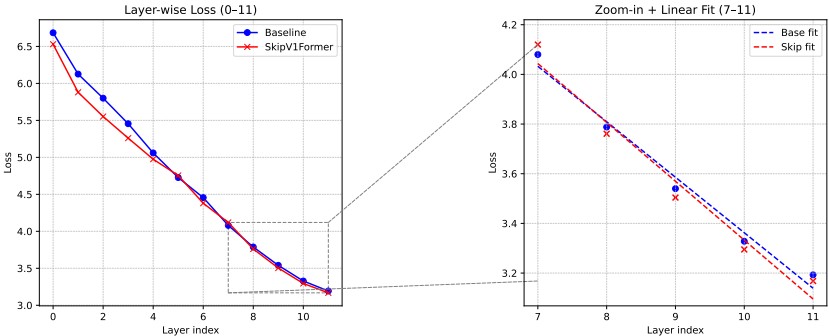

Figure 12: **Layerwise linear probe on a 12-layer GPT2 model.** We train a linear probe on the output of each layer (1–12) and plot validation loss for both the Baseline model and SkipV1Former. On the right, we zoom into layers 7–12 and overlay best-fit lines. The steeper negative slope for SkipV1Former indicates a faster decrease in probe loss—evidence that its cross-layer V reuse strengthens the model's mesa-optimization, yielding more rapidly improving representations at deeper layers.

## C.2  Visualization

To further interpret how cross-layer Value skips in SkipV1Former affect attention patterns and information propagation, we visualize both head–head and token–token similarity matrices at selected layers. These visualizations reveal changes in head diversity and the degree of "oversmoothing" [62, 63] across layers.

Figure 13 shows that SkipV1Former exhibits lower inter-head similarity, indicating higher head diversity. This can be attributed to the injection of uncompressed first-layer Values into deeper layers, which promotes the diversity of Value in deeper layers. Meanwhile, Figure 14 shows that, apart from a few tokens, SkipV1Former reduces the overall token similarity compared to the baseline, alleviating the oversmoothing effect. By supplying deeper layers with original first-layer Values, tokens are less dependent on copying redundant information from neighboring positions, thus preserving more distinct features.

Beyond these similarity analyses, we also provide additional insights into the role of selected heads in SkipV1Former. Prior works have identified previous-token copying heads and induction heads in autoregressive tasks [47]. Specifically, Olsson et al. found that there exists a circuit implemented by two sorts of heads:

- Previous-token copying head: it finds the previous instance of the current token $A$ and the token that came after (call it $B$).

- Induction head: it "completes the pattern" by predicting $B$ after the current $A$.

This characterization is closely related to our theoretical analysis of a 'copying mechanism' in Section 4. We hypothesize that the skipping heads in SkipV1Former play a similar role to previous-token copying heads—leveraging layer 1's complete raw information to facilitate pattern recall—while the remaining heads in layers 2–$L$ operate more like induction heads. In this way, SkipV1Former effectively replaces some of the redundant copying heads in deeper layers and preserves the more functionally important induction heads.

Overall, our visualizations and analysis together suggest that SkipV1Former's cross-layer Value reuse encourages greater head specialization and mitigates representation collapse across tokens—two key factors underpinning its improved representational capacity.

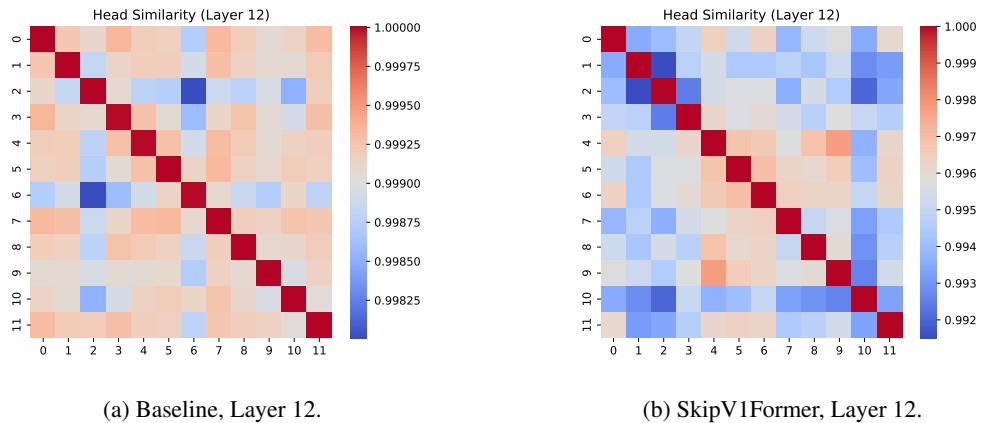

(a) Baseline, Layer 12.              (b) SkipV1Former, Layer 12.

Figure 13: Head–head cosine similarity at layer 12, comparing Baseline and SkipV1Former.

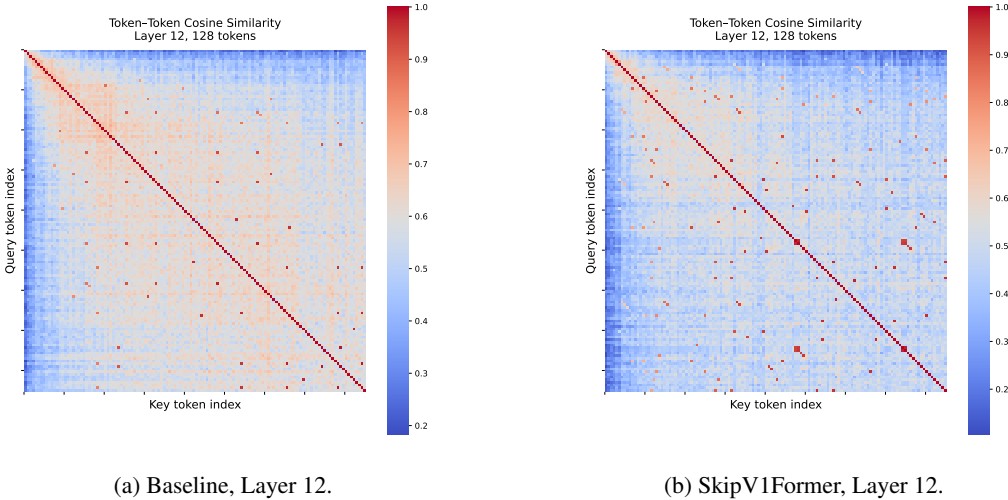

(a) Baseline, Layer 12.              (b) SkipV1Former, Layer 12.

Figure 14: Token–token cosine similarity at layer 12, comparing Baseline and SkipV1Former.

## C.3 Downstream Evaluation

**GPT2-Series.** We assess the performance of our pretrained GPT-2 models across several standard downstream tasks, including HellaSwag, ARC-Challenge, ARC-Easy, PIQA, Winogrande, OBQA, and SciQ. The results are presented in Tables 8 to 11. Notably, SkipV1Former and ResFormer models consistently achieve the highest average accuracy across various model scales, aligning with their performance during pretraining.

**LlaMA-Series.** Similarly, we evaluate the 60M to 1B parameter LLaMA models on downstream tasks such as ARC-Challenge, ARC-Easy, BoolQ, HellaSwag, OBQA, PIQA, RTE, SciQ, and Winogrande. As detailed in Table 12, SkipV1Former consistently outperforms the baseline models, demonstrating improved average accuracy across these tasks.

Table 8: Zero-shot accuracy of six GPT2-125Ms variants across seven benchmark tasks. "Avg." reports the mean accuracy over all tasks. Best performance per column is in bold.

| Model | HellaSwag | ARC-C | ARC-E | PIQA | Winogrande | Obqa | Sciq | Avg. |
|---|---|---|---|---|---|---|---|---|
| Baseline | 27.2 | 17.5 | 40.9 | 60.1 | 51.0 | 14.6 | **67.7** | 39.9 |
| SkipV1Former | **27.3** | 18.0 | 40.9 | 60.1 | **51.5** | 15.0 | 67.6 | **40.1** |
| ResFormer | **27.3** | 18.2 | **42.9** | **60.2** | 50.5 | 14.2 | 67.2 | **40.1** |
| DenseFormer | **27.3** | **18.6** | 40.7 | 59.7 | 51.1 | **15.8** | 65.3 | 39.8 |
| YOCO | 27.2 | 17.6 | 39.4 | 59.6 | 52.0 | 14.4 | 66.2 | 39.5 |
| CLA | 27.2 | 19.1 | 39.9 | 59.4 | 49.3 | 14.4 | 67.0 | 39.5 |

Table 9: Zero-shot accuracy of six GPT2-200Ms variants across seven benchmark tasks. "Avg." reports the mean accuracy over all tasks. Best performance per column is in bold.

| Model | HellaSwag | ARC-C | ARC-E | PIQA | Winogrande | Obqa | Sciq | Avg. |
|---|---|---|---|---|---|---|---|---|
| Baseline | 27.6 | **18.9** | 42.1 | 61.4 | 50.4 | 13.8 | 69.2 | 40.5 |
| SkipV1Former | **28.0** | 17.7 | **43.7** | 60.1 | 49.0 | 14.8 | **72.2** | 40.8 |
| ResFormer | 27.9 | 18.6 | 42.0 | **61.8** | 51.2 | **16.4** | 69.6 | **41.1** |
| DenseFormer | 27.6 | 18.5 | 41.2 | 60.6 | 51.3 | 16.2 | 66.8 | 40.3 |
| YOCO | 27.6 | 18.3 | 41.4 | 60.3 | **51.5** | 16.0 | 69.3 | 40.6 |
| CLA | 27.5 | 17.7 | 40.7 | 60.2 | 50.7 | 15.2 | 66.9 | 39.8 |

Table 10: Zero-shot accuracy of six GPT2-355M variants across seven benchmark tasks. "Avg." reports the mean accuracy over all tasks. Best performance per column is in bold.

| Model | HellaSwag | ARC-C | ARC-E | PIQA | Winogrande | Obqa | Sciq | Avg. |
|---|---|---|---|---|---|---|---|---|
| Baseline | 28.6 | **21.0** | 42.2 | 61.5 | 51.3 | **16.6** | 71.6 | 41.8 |
| SkipV1Former | **29.1** | 18.9 | 43.6 | **62.6** | **52.5** | 15.8 | **73.5** | **42.3** |
| ResFormer | 28.8 | 19.8 | 42.5 | 62.2 | 50.9 | 14.8 | 71.8 | 41.5 |
| DenseFormer | 28.7 | 19.7 | 44.4 | 61.3 | 51.5 | 16.4 | 70.7 | 41.8 |
| YOCO | 28.5 | 18.8 | 42.4 | 61.4 | 51.0 | 15.4 | 68.6 | 40.9 |
| CLA | 28.5 | 19.4 | 41.7 | 61.3 | 51.0 | 15.8 | 69.3 | 41.0 |

## C.4 Alternative Head Injection Strategies

In this section, we investigate different head injection strategies other than our default strategy, which directly concatenates the second-half Value heads of layer 1 to the first half Value heads of layer 2 - $L$. Specifically, we evaluate

- Pooling: Inject the head-wise average of the first half and second half heads in layer 1.
- Dynamic: For the $i$-th layer, inject head $i$ from layer 1 into position $i + H/2$ in a rotating manner.
- Odd/Even: Use layer 1's first-half heads for odd-indexed layers and its second-half heads for even-indexed layers.
- SkipV1 + ResFormer: On top of SkipV1Former, linearly add the first-half heads from layer 1 to the first-half heads of all subsequent layers.

Table 11: Zero-shot accuracy of six GPT2-550Ms variants across seven benchmark tasks. "Avg." reports the mean accuracy over all tasks. Best performance per column is in bold.

| Model | HellaSwag | ARC-C | ARC-E | PIQA | Winogrande | Obqa | Sciq | Avg. |
|---|---|---|---|---|---|---|---|---|
| Baseline | 29.5 | 19.9 | 46.9 | 62.8 | 51.1 | 16.6 | 72.4 | 42.7 |
| SkipV1Former | 29.6 | 18.5 | 46.1 | 62.7 | 51.4 | **17.6** | **73.9** | 42.8 |
| ResFormer | **30.1** | 19.3 | **47.1** | **63.7** | **52.2** | 17.0 | 73.4 | **43.2** |
| DenseFormer | 29.4 | **20.6** | 44.2 | 62.2 | 49.8 | 15.8 | 72.7 | 42.1 |
| YOCO | 29.2 | 19.7 | 45.0 | 62.2 | 50.5 | 15.0 | 71.5 | 41.9 |
| CLA | 28.8 | 19.0 | 43.2 | 62.1 | 51.5 | 15.8 | 72.1 | 41.8 |

Table 12: Test accuracies (%) of 60M - 1B scale models on downstream tasks, with overall average.

| Model | ARC-C | ARC-E | BoolQ | Hella | OBQA | PIQA | RTE | SciQ | Winogr | Avg |
|---|---|---|---|---|---|---|---|---|---|---|
| Baseline-1B | 19.3 | **44.5** | **60.2** | 31.0 | 17.2 | 66.3 | **53.4** | 71.6 | **52.0** | 46.2 |
| SkipV1Former-1B | 19.3 | 44.1 | 59.6 | **31.2** | **17.4** | **66.9** | 52.3 | **73.8** | 50.9 | 46.2 |
| Baseline-350M | 19.7 | **45.1** | **59.4** | 30.1 | 16.6 | 64.3 | 50.9 | 71.8 | 51.4 | 45.5 |
| SkipV1Former-350M | **20.8** | 44.9 | 53.0 | **30.9** | **18.0** | **66.3** | **52.7** | **74.5** | **51.5** | **45.8** |
| Baseline-130M | **19.4** | 37.1 | 50.3 | 27.2 | **16.0** | 60.9 | 49.8 | 65.6 | **52.0** | 42.0 |
| SkipV1Former-130M | 17.8 | **39.3** | **61.3** | **27.7** | 14.8 | **61.9** | **53.8** | **68.5** | 49.8 | **43.9** |
| Baseline-60M | **17.8** | 33.0 | 45.3 | 26.2 | **13.6** | **59.3** | **59.2** | 58.8 | 50.9 | 40.5 |
| SkipV1Former-60M | 16.7 | 33.0 | **60.1** | **26.6** | 12.8 | 58.9 | 52.3 | **64.3** | **51.0** | **41.7** |

Table 13 shows that there is no substantial improvement of these strategies over the simple second half scheme. Moreover, our default injection strategy selects preserves a consistent head ordering across layers, which is more hardware friendly than discontinuous schemes like random head selection [45]. In all, simply injecting only the second half heads has achieved certain model performance improvement while being more hardware-friendly.

## C.5 Skip Connections with both K and V

In this section, we investigate methods to further reduce the Key-Value (KV) cache size by using skip connections on both Keys and Values. A straightforward approach is to reuse the first layer's K and V in all subsequent layers via skip connections. We term this architecture as SkipKV1Former. Additionally, we propose SkipV1-YOCO, which combines the Keys sharing mechanism in YOCO [15] (see Appendix B.1) with our skip connection approach for the first Value in SkipV1Former. Specifically, the first $\frac{L}{2}$ layers of SKipV1-YOCO are the same as those in SkipV1Former, while the rest $\frac{L}{2}$ layers computes attention by:

$$\text{Attn}(X) = X + \sum_{h=1}^{H'} W_O^h V^h \text{softmax}\left(\left(K_{L/2}^h\right)^\top Q^h\right) + \sum_{H'+1}^{H} W_O^h V_1^h \text{softmax}\left(\left(K_{L/2}^h\right)^\top Q^h\right).$$

Both SkipKV1Former and SkipV1-YOCO reduce the KV cache size for approximately 50%. We pretrain SkipKV1Former and SkipV1-YOCO on GPT2-125M to GPT2-355M using the same hyperparameter settings as in our GPT2 pretraining experiments.

Figure 15 illustrates the performance of these models across three different scales. Both SkipKV1Former and SkipV1-YOCO are outperformed by SkipV1Former, indicating that skip connections on Keys may lead to performance degradation. On the other hand, SkipV1-YOCO shows only a slight performance drop compared to SkipV1Former and still outperforms the baseline model, whereas SkipKV1Former exhibits inferior performance even compared to the baseline model. The relatively strong performance of SkipV1-YOCO suggests that selective reuse of Keys (as in YOCO) may mitigate performance degradation, pointing to a promising direction for further KV cache reduction.

| Method | SkipV1 | Pooling | Dynamic | Odd/Even | SkipV1 + ResFormer |
|---|---|---|---|---|---|
| Val. Loss | 2.885 | 2.881 | 2.885 | 2.883 | 2.889 |

Table 13: Validation loss of different head injection strategies on LLaMA-350M model.

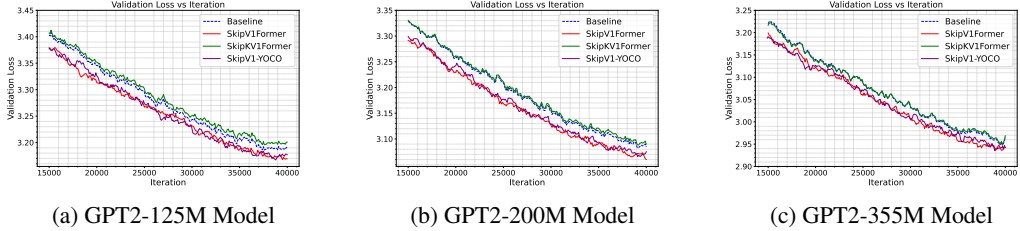

|                          |                         |                         |
|:------------------------:|:-----------------------:|:-----------------------:|
| (a) GPT2-125M Model      | (b) GPT2-200M Model     | (c) GPT2-355M Model     |

Figure 15: Validation loss over training iterations for GPT2-125M, GPT2-200M, and GPT2-355M models, comparing SkipV1-YOCO and SkipKV1Former with the MHA baseline (blue line) and SkipV1Former (red line).

## C.6 Vision Transformer

Although the design and theoretical analysis of SkipV1Former target towards auto-regressive tasks, in this section we conduct experiments on vision tasks for completeness. We train a ViT-Tiny model [64] on CIFAR-10 and CIFAR-100 from scratch to preliminarily test the performance of SkipV1Former on vision tasks. Specifically, our Vit-Tiny model has 12 layers and 3 heads per layer, with a hidden size 192 and an MLP of dimension 768. The total number of parameters is 5.5M. Though CIFAR datasets are generally considered too small for ViT [65], our comparisons between SKipV1Former and the baseline model are controlled and fair. We train the models using AdamW with $\beta_1 = 0.9, \beta_2 = 0.999$ and weight decay 0.1. The peak learning rate is 1e-3, with a 10% warmup ratio and cosine annealing that decays to 10% of the peak. We use a batch size of 1024 and train for 15000 steps, i.e., 300 epochs.

Figure 16 shows test accuracy over training on CIFAR-10 and CIFAR-100. SkipV1Former does not outperform the standard MHA Transformer. This indirectly supports our theoretical analysis, which is rooted in the mesa-optimization behavior observed in trained Transformers for auto-regressive tasks. In non-autoregressive tasks, accessing information from the first layer cannot preserve the causal structure between samples and labels. Instead, the raw features from the first layer are less processed and may disrupt the hierarchical feature extraction that vision Transformers rely on, leading to degraded performance when being injected to deeper layers. Designing an effective skip connection scheme for vision Transformers may require a deeper understanding of how information flows through the learned architecture.

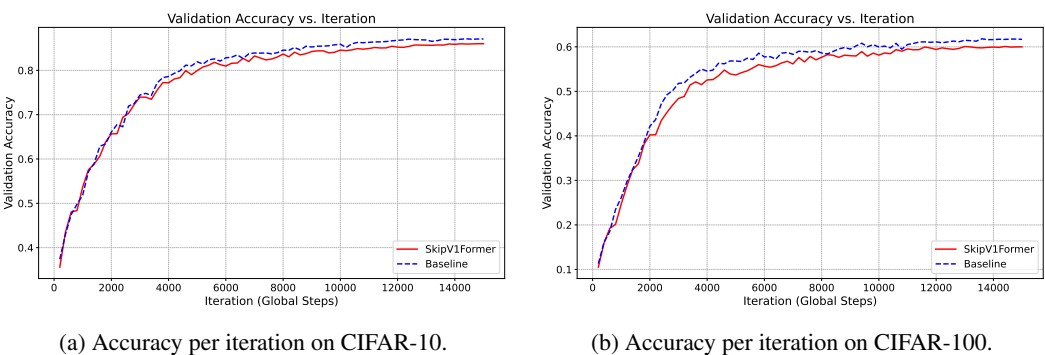

|                                         |                                          |
|:---------------------------------------:|:----------------------------------------:|
| (a) Accuracy per iteration on CIFAR-10. | (b) Accuracy per iteration on CIFAR-100. |

Figure 16: Accuracy per iteration of the baseline versus SkipV1Former on a Vision Transformer: results on CIFAR-10 (left) and CIFAR-100 (right).

