# OpenReview forum: "Improving Model Representation and Reducing KV Cache via Skip Connections with First Value Heads"
_NeurIPS.cc/2025/Conference — NeurIPS 2025 poster_

### Official Review · Reviewer_oBgN · 2025-06-30

**Clarity:** 2
**Significance:** 3
**Originality:** 3
**Rating:** 4
**Confidence:** 3

**Summary:**

They provide a method called SkipV1Former, which allows each layer to reuse half of its Value heads from the very first layer. Their method utilizes the property of self-attention can save KV cache and improve GPU memory usage, and at the same time not degrade the performance.

**Questions:**

1. Based on Figures 14a and 14b, it seems that the larger model gets improved less than the smaller model with SkipV1Former. Also, in Figure 5a, it seems the larger model also has a worse perplexity improvement. I wonder if 50% is still the best for the larger models (1B, 3B)
2. What is the reason for the intuition that shallow layers would be more effective than deeper layers.

**Ethical Concerns:**

["NO or VERY MINOR ethics concerns only"]

**Final Justification:**

The author addressed most of my concerns. I have increased my score.

**Limitations:**

1. The improvement for the larger model is a bit small. How they choose how much of the Value to be replaced seems to only depend on the ablation study of GPT2-355M.

**Quality:**

3

**Strengths And Weaknesses:**

## Strengths
1. The motivation is clear. The authors clearly state the drawbacks of the two categories of skip-connection techniques. Their designed method solves those two problems simultaneously.
2. Their proposed method is flexible, which can also be combined with other KV-saving approaches to attain greater savings.

## Weaknesses
1. The effectiveness of the method on larger models is limited. Based on Table 1 and Figure 5a, the absolute value of the improvement of the test accuracy and the perplexity seems to be a bit small. It would be better if the author could include a significance test.
2. It is unclear why the method seems to be more effective for larger models. In the ablation study (Section 6.5), different skip-head ratios are only tested on GPT2-355M, which is relatively small compared to LLaMa models. Therefore, the reasons for choosing 50% seem not very convincing. It would be better if authors could include ablation studies for larger models.
3. The author discusses why the first layer matters in Section 4. In line 136, the authors argue that "it seems natural that injecting shallow-layer features into deeper layers enriches representation." However, I am confused why this is natural. It would be better if the author could provide some references. In line 137, it points to an ablation study, which only studies the first three layers. It would be better if the author could provide a study on deeper layers to prove their claim.
4. For clarity, it is confusing how the proposed method improved efficiency. Discussion of how it improved prefill time, throughput, and GPU memory is not emphasized in the main text. Most of them are in the appendix. I would suggest the author move those discussion back to main text

---

> ### Author Rebuttal · Authors · 2025-07-30
>
> Thank you for your careful review and thoughtful questions and comments. Please see our response below.
> > The effectiveness of the method on larger models is limited. Based on Table 1 and Figure 5a, the absolute value of the improvement of the test accuracy and the perplexity seems to be a bit small. It would be better if the author could include a significance test.
>
> Thank you for your feedback. We deem that the main contribution of SkipV1Former is the **simultaneous improvement in improving accuracy and reducing KV cache**. While SkipV1Former’s absolute gains in loss and perplexity may appear modest on larger models, it consistently reduces KV cache by ~25% and delivers reliable quality improvements across scales.
>
> To address your concerns, we retrain 1B SkipV1Former and baseline model using another random seed (The original random seed is 42. Due to our limited computation resources, now we are only able to run one set of experiment). It is shown in the following table that SkipV1Former outperforms the baseline by a similar margin. In addition, we also conduct experiments on TinyLlaMA 1.1B and find similar results. Please refer to our response to Weakness 2 to Reviewer 8cpz. We will complete the significance test on 1B models in the final version of the paper.
>
> **Table: Validation loss of 1B-LlaMA models under different random seed.**
>
> | Model | Baseline (seed 42) | SkipV1 (seed 42) | Baseline (seed 100) | SkipV1 (seed 100) |
> |:-----:|:----------:|:----------:|:-----------:|:-----------:|
> | Val. Loss  |    2.723   |    2.711   |    2.753    |    2.743    |
>
>
> > It is unclear why the method seems to be more effective for larger models. In the ablation study (Section 6.5), different skip-head ratios are only tested on GPT2-355M, which is relatively small compared to LLaMa models. Therefore, the reasons for choosing 50% seem not very convincing. It would be better if authors could include ablation studies for larger models.
>
> For SkipV1Former, we choose a 50% skip head ratio to balance memory savings and model performance. Injecting fewer than 50% of the first layer heads (e.g., 25%) yields limited improvement in memory usage, whereas injecting all heads—as in SVFormer (see Ref. [25]. SVFormer is a variant of ResFormer, which happens to be an extreme variant of SkipV1Former that choose 100% skip-head ratio)—introduces significant performance degradation compared to the baseline.
>
> To further validate this trade off, we conducted a preliminary ablation on the 1B parameter LLaMA model.
> It can be seen that, although a 75% skip ratio does provide additional memory reduction, it incurs a noticeable drop in accuracy. Conversely, a 25% skip ratio fails to outperform the 50% setting. These results suggest that 50% may represents a near-optimal tradeoff between memory efficiency and model quality. We are currently extending these experiments to explore additional skip ratios, and will include the complete ablation study in the final version of the paper. We hope our additional results can address your concerns.
>
> **Table: Validation loss of 1B-LlaMA models with different skip-head ratios.**
>
> | Skip Head Ratio/Model | 25% | 50% (default) | 75% | baseline |
> |:-----:|:----------:|:----------:|:-----------:|:-----------:|
> | Val. Loss  |    2.716   |    2.711   |    2.717    |    2.723    |
>
>
> > The author discusses why the first layer matters in Section 4. In line 136, the authors argue that "it seems natural that injecting shallow-layer features into deeper layers enriches representation." However, I am confused why this is natural. It would be better if the author could provide some references. In line 137, it points to an ablation study, which only studies the first three layers. It would be better if the author could provide a study on deeper layers to prove their claim.
>
>
> Thank you for pointing out this ambiguity. The comparison here is between **SkipV1Former**, which injects half of layer 1’s heads into every deeper layer, and **SkipV0** in Fig. 8(b), Section 6.5, which simply drops those heads from layers 2–L without any first‐layer reuse. Our intent is to highlight the importance of the first layer. Specifically, the attention of SkipV1 can be expressed as the equation in line 122 and the attention of SkipV0 is $\text{Attn}(X) = X + \sum_{h=1}^{H’} W_O^h V^h \text{softmax}((K^h)^{\top}Q^h)$. Therefore, SkipV1Former’s attention can express SkipV0’s output (e.g., $W^h_O = \textbf{0}$ for $h=H'+1,\cdots,H$), implying an improved representation capability. This is why we say `naturally enriches representation’. Similar injection/concatenation techniques have been used in DenseNet [1] to improve representation. We will modify the statement of this paragraph in the final version of the paper.
>
> To validate our claim beyond the first two layers (Fig. 8(b)), we extended the ablation studies at layer 3, 4, and 6 (Since inserting Value heads at deeper layer can only cut heads from that layer onward, leading to less V cache and parameter savings, we stop at a layer 6). As shown in the tale below, injecting heads of layer 1 consistently outperforms all other variants, confirming the unique importance of first layer head reuse. We will include these detailed results and discussions in the final version of the paper.
>
> **Table: Relative validation loss for cross-layer V at different layers.**
>
> | Model | SkipV1 | SkipV0 | SkipV2 | SkipV3 | SkipV4 | SkipV6 |
> |:-----:|:----------:|:----------:|:-----------:|:-----------:|:-----------:|:-----------:|
> | Relative Val. Loss  |    0   |      +0.042    |     +0.025    |     +0.025    |     +0.019   |  + 0.041  |
>
>
> [1] Densely Connected Convolutional Networks
>
> > For clarity, it is confusing how the proposed method improved efficiency. Discussion of how it improved prefill time, throughput, and GPU memory is not emphasized in the main text. Most of them are in the appendix. I would suggest the author move those discussions back to main text.
>
> Thank you for your valuable suggestions. As shown in line 122-125 and Figure 1 (Right), SkipV1Former cuts half of the $W_V$ parameters from layer 2 - L compared to standard Transformer. This removal directly reduces the **number of heads of every V cache** (aka., the embedding dimension of V) from layer 2 - L and also the **corresponding matmul computations** of $W_V$, resulting in improvement on GPU memory, prefill time and throughput. Due to the limit of pages, we had to move some discussions on prefill time and throughput to the Appendix in the original manuscript, as the main contribution of SkipV1Former lies in performance boosting and KV cache savings. We will modify our discussions on how SkipV1Former improves inference and memory and move the discussions in the Appendix back to the main text.
>
>
> > Based on Figures 14a and 14b, it seems that the larger model gets improved less than the smaller model with SkipV1Former. Also, in Figure 5a, it seems the larger model also has a worse perplexity improvement. I wonder if 50% is still the best for the larger models (1B, 3B)
>
> Please see our response for Weakness 2 above. We think that 50% is a near-optimal ratio for the tradeoff between performance and memory savings. We extend our ablation on skip-head ratio on 1B models and find that 50% is still the best (please see our response to Weakness 2 above). We will conduct more ablation studies on larger models in the final version of the paper.
>
>
> > What is the reason for the intuition that shallow layers would be more effective than deeper layers.
>
> Thank you for your feedback. We first clarify that we do **not** claim that shallow layers are **strictly** more effective than deep layers as we only inject half of the heads of layer 1 instead of all heads nor heads from other shallow layers. In Section 4 and Appendix A, we present some discussions and theoretical analysis to show why injecting certain information from shallow layers directly can be beneficial. In short, trained Transformer performs `copying mechanism’ in shallow layers in auto-regressive tasks, which incurs certain information loss in the propagation and thus slower down the mesa-optimization. By contrast, by directly injecting half of layer 1’s heads, SkipV1Former restores access to the raw information of the input, enabling deeper layers to perform mesa-optimization with more complete information.
> Moreover, deeper layers themselves tend to develop certain redundant representations in standard Transformer (e.g., see Ref. [15]). Thus, removing half heads from layer 2-L and concatenate the rest with the second half of the layer 1 brings gains. We will refine this discussion in the final paper.
>
> > The improvement for the larger model is a bit small. How they choose how much of the Value to be replaced seems to only depend on the ablation study of GPT2-355M.
>
> Please see our response to Weakness 1 and 2 above. Importantly, SkipV1Former consistently reduces KV cache by ~25% while still delivering accuracy gains compared to the standard Transformer—even at larger model scales.

---

> > ### Comment · Reviewer_oBgN · 2025-08-04
> >
> > Thank you for your response. I appreciate the additional experiments with different random seeds and skip-head ratios, as well as your efforts to test skips at different layers.
> >
> > However, I still have a few questions, particularly regarding the skip ratios. Based on the reported validation loss, the 25%, 50%, and 75% skip ratios appear to yield similar results. You mention that a 75% ratio leads to a noticeable drop in accuracy, but it’s unclear how large that drop is. I wonder what the accuracy difference is.

---

> > > ### Author Response · Authors · 2025-08-05
> > >
> > > We sincerely thank you for taking the time to review our response and your follow-up question. Below, we quantify the relative validation-loss and perplexity improvements over the baseline:
> > >
> > > **Table: Relative gains for different skip-head ratios compared with baseline.**
> > >
> > > | Skip Ratio | GPT-2 355M Rel. Loss Δ | GPT-2 355M PPL Δ (%) | LlaMA 1B Rel. Loss Δ | LlaMA 1B PPL Δ (%) |
> > > |:----------:|:----------------------:|:----------------------:|:--------------:|:-----------------------:|
> > > | **50%**    | –0.024 |   18.91 (-2.4%)     | –0.012         | 15.04 (–1.2%)          |
> > > | **75%**    | –0.013 |   19.15 (-1.2%)     | –0.006         | 15.13 (–0.6%)          |
> > >
> > > Across both model scales, the 75% ratio recovers about 50% of the improvement achieved by the 50% ratio setup, which we view as a drop in benefit.
> > >
> > > That said, the 75% skip ratio does offer greater memory savings while maintaining performance comparable to the baseline, which may be attractive in highly memory-constrained environments. In addition to larger skip head ratio, SkipV1Former can be combined with YOCO (Appendix C.4) to achieve further KV cache savings while preserving baseline compatibility. We will include these detailed quantifications and discussions of skip-head ratios in the final manuscript.

---

> > > > ### Comment · Reviewer_oBgN · 2025-08-05
> > > >
> > > > Thank you for your response. Your answer addressed my concerns and I will take this into consideration when revising my score.

---

> > > > > ### Author Response · Authors · 2025-08-05
> > > > > **Thank you**
> > > > >
> > > > > Thank you again for your valuable feedback. We will continue improving our paper.

---

### Official Review · Reviewer_8cpz · 2025-07-01

**Clarity:** 2
**Significance:** 2
**Originality:** 2
**Rating:** 4
**Confidence:** 5

**Summary:**

This paper proposed a KV reduction/sharing technique which injects half of the V of first layer into later layers so that the computation and cache of later V layers will be reduced.  This technique requires retraining from scratch or reduced fine-tuning (uptraining) from existing MHA checkpoints in order to achieve comparable or slightly better performance than baseline models.  Furthermore, the author tested the compatibility of the proposed method with other popular KV cache reduction works, such as GQA and MLA.  A theoretical analysis was also provided in Appendix to give insights on how and when this method may or may not work, such as the case of ViT.

**Questions:**

see Weakness above

**Ethical Concerns:**

["NO or VERY MINOR ethics concerns only"]

**Final Justification:**

Concerns about the selection of heads to be injected is addressed with new experimental data. Question regarding GQA with group-of-4 configuration is partly address with new data using smaller models. Hence the adjustment of the rating.

**Limitations:**

yes

**Quality:**

3

**Strengths And Weaknesses:**

Strength:

1.	A good amount of decent works, including theoretical analysis, experiments with relevant techniques, tested on a range of model sizes, and the comprehensive details in Appendix.
2.	Method seems simple enough but yet works reasonably fine.


Weakness:

1.	a tad counter-intuitive

    - based on author's analysis, the way this method works is to inject "raw" information from V1 into later layers which may enhance model's representation capability. In other words, the existing skip connections (residual path) in transformer architecture, i.e., after attention block and MLP block, are not sufficient. Intuitively the "raw" information could be mostly preserved through an identity-like head and the relative mixing ratio could be modulated by the magnitude of last Linear layer in each block before adding the residual, e.g. "o_proj" and "down_proj" in Llama. These should have been optimized during model training, therefore, the need for additional direct injection, especially only partial "raw" information, doesn't seem necessary.
    - Even if we need to do a direct injection, why is 50% a lot better than others, (Fig 8a)? and why only inject the second half of V1, wouldn't some information still get lost this way? maybe something like an average pool of every 2 heads, as used by author's uptraining initialization, would make more sense? Also will it make any differences to inject the first half, or even/odd, or randomly selected heads? Maybe the author could provide more experimental data on this part to reinforce the hypothesis?
    - It would be nice to include this discussion in the manuscript/appendix so that the readers could have a clearer picture of how this method works.

2.	generalization with newer architectures

The GQA example given in the manuscript is based on Llama (seems to be version 1) with group-of-two scheme (Appendix B2.) However, in newer Llama (version 2 70b or version 3), GQA is applied with group-of-4 configuration, i.e. K and V has 8 heads instead of 32. It would be more beneficial to demonstrate with the more recent and realistic settings. As many readers may assume that when the redundancy is reduced in such GQA scheme, SkipV1Former method might not be able to maintain 50% injection from V1, hence, the benefit of this method could be reduced.

3.	other minor suggestions
    - the parameter count reduction from SkipV1Former in Table 2 seems to be too small. Almost all the parameters in the V layers are halved, but parameter counts is only reduced by ~6M out of ~335M? maybe add a few sentences in that paragraph or in Appendix to demonstrate the calculation?
    - Appendix C5 provides some useful inference speed test results. Even though the throughput at decoding stage is not improved as expected, it would be nice to include a little more information regarding the preliminary analysis of the source of the overhead, e.g. the author seems to have performed a profiling and suggests that the stitching of half of the V1 into later Vs is causing the slow-down. However, it's unclear how this stitching is different from normal V cache retrieval and concatination. Maybe a little more clarification would help the readers to understand if this is a kernel optimization problem or something else.

---

> ### Author Rebuttal · Authors · 2025-07-30
>
> Thank you very much for your careful review and thoughtful questions and comments. Please see our response below.
>
> > a tad counter-intuitive… The need for additional direct injection, especially only partial "raw" information, doesn't seem necessary
>
> Thank you for your feedback. We found that some of the terminology such as “identity like head” is a bit confusing, so let us first restate our understanding of your concern: you are asking whether the existing residual paths already suffice to preserve “raw” first layer information in deeper layers, making our cross layer head injection unnecessary. In fact, there is an essential difference between the residual connections in standard model and our first-layer head injection:
> - In a standard Transformer, each layer’s input $X^k$ is linearly transformed by the attention and MLP sublayers and then **added back** to the original $X^k$ via the residual connection. This process **fuses only adjacent layers** and does not establish any direct Value to Value linkage between layer 1 and layers 2–L.
> - By contrast, SkipV1Former explicitly **concatenates** half of the Value heads from layer 1 with the Value heads of every subsequent layer (note that SkipV1Former also utilizes the conventional residual connections).
>
> As shown in Ref. [14] (Fig. 2) and Section 4, even with residual connections, standard Transformer exhibit a “copying mechanism” in early layers that incurs information loss. On the other hand, direct head injection preserves these raw features, enabling deeper layers to optimize with more complete information. Empirically, across our experiments on GPT and LlaMA models, SkipV1Former consistently improves perplexity and downstream metrics over the baseline. This confirms that first layer head injection is not redundant but rather a distinct and beneficial mechanism beyond conventional residual connections.
>
> > Even if we need to do a direct injection, why is 50% a lot better than others, (Fig 8a)?
>
> Thank you for your question. The choice of the skip ratio is based on both model performance and memory savings. Since we maintain the same number of heads of the concatenated $V$, a larger ratio of skipping heads implies a larger reduction on the embedding dimension of $W_V$ from layer 2-L. Skipping fewer than 50% heads yields less memory savings, while skipping 100% heads causes a clear drop in accuracy (see SVFormer in Ref. [25]). Intuitively, skipping too few heads results in certain information loss caused by `copying mechanism’ for deeper layers, whereas skipping too many heads hinders the capability of deeper layer to process the full data flow. Empirically, a 50% skip ratio strikes a good balance between memory savings and model quality as shown in Fig. 8a, supporting this design choice. We are extending these ablation studies to larger LLaMA models and will include the full results in the final paper (please see also our detailed response to Weakness 1 to Reviewer oBgN).
>
>
> > And why only inject the second half of V1, wouldn't some information still get lost this way? … Maybe the author could provide more experimental data on this part to reinforce the hypothesis?
>
> Thank you for your valuable question. We choose to inject heads $H/2+1$ to $H$ from layer 1 because **continuous, block-wise selection** preserves a consistent head ordering across layers, which is more hardware friendly than discontinuous schemes like random head selection [1].
>
> In early experiments, we also tried injecting the first half of layer 1’s heads via dynamic strategies and ResFormer style mixing, but these variants delivered only marginal gains.
> To explore further, we conducted an extended ablation on LLaMA 350M with several head injection strategies:
>
> - **Pooling**: Injecting the head-wise average of the first half and second half heads in layer 1.
> - **Dynamic**: Injecting head $i$ to $i + H/2$ in layer 1 for the $i$-th layer in a rotating manner.
> - **Odd/Even**: Injecting layer 1’s first half heads for odd-indexed layers and the second half heads for even-indexed layers.
> - **SkipV1 + ResFormer**: Linearly add the first half heads to all the first half heads in subsequent layers upon SkipV1Former.
>
> As the table shows, we found no substantial improvement of these strategies over the simple second half scheme. In short, simply injecting only the second half heads has achieved certain model performance and may be more hardware-friendly.  We will incorporate this discussion and the full experimental results in the final manuscript.
>
> **Table: Validation loss of different head injection strategies on LLaMA-350M model.**
>
> | Method | SkipV1 | Pooling | Dynamic | Odd/Even | SkipV1 + ResFormer |
> |:-----:|:----------:|:----------:|:-----------:|:-----------:|:-----------:|
> | Val. Loss  |    2.885   |    2.881   |    2.885    |    2.883    |    2.889   |
>
>
>
> [1] Native Sparse Attention: Hardware-Aligned and Natively Trainable Sparse Attention
>
> > generalization with newer architectures: It would be more beneficial to demonstrate with the more recent and realistic settings
>
> Thank you for your valuable suggestion. We agree that it will be better to experiment on more advanced models such as GQA-4 Transformers. However, due to our limited computing resources, we now only able to conduct the uptraining experiments on relatively small models.
> Specifically, we compare the performance of SkipV1Former and the baseline model utilizing GQA with group-of-4 configuration on TinyLlama 1.1B models. We also test the performance on 315M and 125M TinyLlama models by modifying the configurations in the same manner as Llama 130M/350M with Llama 1B.
> It is shown in the following table that SkipV1Former still outperforms the baseline model by a similar margin as that in Table 2 for GQA-2 models. We hope these results can address your concerns and we will include them in the final version of the paper.
>
> **Table: Validation loss across 3 TinyLlaMA model sizes (125M, 315M, 1.1B).**
>
> | Model | T-LlaMA (1.1B) | T-LlaMA-SkipV1 (1.1B) | T-LlaMA (315M) | T-LlaMA-SkipV1 (315M) | T-LlaMA (125M) | T-LlaMA-SkipV1 (125M) |
> |:-----:|:----------:|:----------:|:-----------:|:-----------:|:-----------:|:-----------:|
> | Val. Loss  |    2.750   |    2.742   |    2.936    |    2.916    |     3.420    |     3.370    |
>
>
> > The parameter count reduction from SkipV1Former in Table 2 seems to be too small.
>
> Thank you for your feedback. We provide the detailed calculations as follows.
>
> - GQA model: the number of layers is 24, the total number of heads (of Queries) is 16, and the dimension of each head is 64, implying that $W_Q \in \mathbb{R}^{1024 \times 1024}$. Since it share the same keys and values between every two Query heads, the number of heads of Keys and Values is 8, implying that $W_V\in \mathbb{R}^{1024\times 512}$. As SkipV1Former reduces $W_V$ parameters by half from layer 2 - L, the reduced number of parameters is $1024\times 256 \times 23 = 6.03$M.
> - MLA model: the number of layers, total heads and dimension of value head is the same as those of GQA model. The kv_lora_rank, aka, $d_c$ in line 661 is set as 256 following the shrinking ratio of deepseek-v3 model. SkipV1Former-MLA cuts $d_c$ to 128 from layer 2 - L, resulting in the reduced number of parameters as $1024 \times 128 \times 23 = 3.02$M.
>
> Note that the reductions on parameter count is a by-product of our skip connection schemes, and the main contribution of SkipV1Former lies in performance and KV-cache savings. We will include all these calculations as well as those of the total number of parameters in Appendix in the final version of the paper.
>
> > However, it's unclear how this stitching is different from normal V cache retrieval and concatenation. Maybe a little more clarification would help the readers to understand if this is a kernel optimization problem or something else.
>
> Thank you for your valuable suggestion. We clarify this as follows: during decoding, for each new token $x_t$,
> - Standard Transformer: it extends the cached V tensor $V[1:t-1] \in \mathbb{R}^{bs\times H\times (t-1) \times d}$ of token $x_1$-$x_{t-1}$ by appending $x_t$’s Value along the **seq_len** dimension, yielding $ V[1:t-1]$.
> - SkipV1Former: it caches Head 1 to H/2 from layer 1 to L (denoted by $V^{l, 1:H/2}[1:t-1]$ for $l=1,\cdots, L$) and the second half heads of layer 1 (denoted by $V^{1, H/2+1:H}[1:t-1]$). When computing attention for $x_t$, it first concatenates $V^{l, 1:H/2}[1:t-1]$ and $V^{1, H/2+1:H}[1:t-1]$ to those heads of $x_t$ along **seq_len** dimension, yielding $V^{l, 1:H/2}[1:t]$ and $V^{1, H/2+1:H}[1:t]$. Then it concatenates these two buffers along the **head** dimension, implementing the cross-layer skip connections.
>
> To better illustrate this, we will include a schematic diagram in the revised Appendix.
>
> Additionally, we have updated our throughput benchmarks at batch sizes 8 and 16, which are more representative of real world inference. Under these settings, SkipV1Former’s decoding throughput matchess or slightly exceeds the baseline (±5%), rather than the average 5% slowdown reported in Appendix C.5. Please see our response to Q3 to Reviewer t3Ns for details.
>
> Since SkipV1Former reduces V projection matmuls by nearly 50%, we think that the throughput of can be further improved through techniques like kernel optimization (e.g., fused concat). We leave this exploration to future work and we will include the discussions in the final version of the paper.

---

> > ### Comment · Area_Chair_6yoZ · 2025-08-05
> > **Discussion**
> >
> > Dear Reviewer 8cpz,
> >
> > The authors have responded to your concerns. How does their response change your view of the paper? If it does not, please clarify what the authors can do to address your concerns. If it does, please consider adjusting your score based on their response.
> >
> > Your AC

---

> > ### Comment · Reviewer_8cpz · 2025-08-06
> >
> > Thanks for the additional information and experimental data. My concerns are mostly addressed. I will adjust my rating accordingly.

---

> > > ### Author Response · Authors · 2025-08-07
> > > **Thank you**
> > >
> > > Thank you again for your valuable feedback. We will continue improving our paper.

---

### Official Review · Reviewer_t3Ns · 2025-07-02

**Clarity:** 3
**Significance:** 3
**Originality:** 3
**Rating:** 5
**Confidence:** 4

**Summary:**

SkipV1Former introduces a simple yet powerful twist to Transformers: starting with the second block, half of every layer’s value heads are borrowed directly from the first layer. This “Value-1 skip” trims both value-projection parameters and the value cache by roughly 50 %, which translates to 25% of total KV-Cache memory saving. Converting an existing checkpoint costs only 10–15 % of the original pre-training compute, and the technique is also able to stack cleanly with methods like Group-Query Attention or YOCO for enabling more memory-efficient inference.

**Questions:**

Q1: How are the heads selected in each layer? What about dynamical selection for different layers?

Q2: Can the author provide some insights on the selected heads? Some other works have shown that different kv_heads play different roles. Are the selected value heads for replacement the relatively redundant ones?

Q3: What is the batch size for measuring the throughputs in Fig. 14 (b)?

Q4: If lowering the compression rate, is SkipV1Former able to operate in a training-free fashion? Some related works, such as MiniCache [1] and xKV [2], have uncovered the opportunity in enabling cross-layer KV-Cache sharing in a post-training fashion. Can authors provide some discussion on this?

[1] MiniCache: KV Cache Compression in Depth Dimension for Large Language Models

[2] xKV: Cross-Layer SVD for KV-Cache Compression

**Ethical Concerns:**

["NO or VERY MINOR ethics concerns only"]

**Final Justification:**

The SkipV1Former provides a compelling architectural insight by introducing skip connections from the Value projections of the first layer directly to later layers. This design is supported by solid theoretical derivations and justifications. Empirically, the model demonstrates improved perplexity compared to baseline methods when trained from scratch, highlighting its effectiveness. While the gains in computational efficiency are modest, the conceptual novelty has the potential to inspire future work in both system-level optimizations and architectural innovations.

**Limitations:**

yes

**Quality:**

4

**Strengths And Weaknesses:**

## Strength:
1. Interesting insights and good mathematical analysis for why skipping the first layer works, I really like this part in Section 4
2. Seamlessly combined with various KV cache efficient architectures, gqa, mla, etc. Proves this method is versatile.
3. Experiments are sufficient, from pretraining to uptraining and memory analysis are good, in combination with other works, YOCO.

## Weakness:
1. Limited Efficiency Gain: The benefit of SkipV1Former mainly lies in memory saving for throughputs/latency; the proposed methods only bring marginal improvements.
2. Insufficient explanation on how heads are selected. It seems that you just keep the indices 0,1,2..., H/2 for the following layers.
3. Some missed related works.
    + At CLLA [1], the proposal of sharing the latent cache of MLA across different layers has been explored. I also suggested including this paper in the discussion.

[1] Lossless KV Cache Compression to 2%

---

> ### Author Rebuttal · Authors · 2025-07-30
>
> Thank you for your careful review and thoughtful questions and comments. Please see our response below.
>
> > Limited Efficiency Gain: The benefit of SkipV1Former mainly lies in memory saving for throughputs/latency; the proposed methods only bring marginal improvements.
>
> We view SkipV1Former’s core contribution as the **combined gains** in accuracy and ~25% KV‑cache reduction. We acknowledge that throughput and latency improvements are modest. On the other hand, many KV-cache saving methods—such as SVD based compression— often require extra computation, which degrades throughputs. It is an interesting direction to further improve the inference efficiency of SkipV1Former. Please see our response to Question 3 below for more discussions on throughputs and latency.
>
> > Insufficient explanation on how heads are selected. It seems that you just keep the indices 0,1,2..., H/2 for the following layers.
>
> We use a **fixed deterministic selection** strategy: for layers 2-L we always keep heads 0…H/2 and concatenate them with heads H/2+1 … H from layer 1. This continuity ensures a stable and aligned head ordering across layers, which is more hardware-friendly compared to other discontinuous methods [1] such as random selection. We will clarify this choice and include the core code implementation in the Appendix.
>
>
> [1] Native Sparse Attention: Hardware-Aligned and Natively Trainable Sparse Attention
>
> > Some missed related works. At CLLA [1], the proposal of sharing the latent cache of MLA across different layers has been explored. I also suggested including this paper in the discussion.
>
> Thank you for highlighting CLLA. We will cite and discuss its cross‑layer MLA cache sharing in the final version of our paper.
>
> > Q1: How are the heads selected in each layer? What about dynamical selection for different layers?
>
> Please refer to our “fixed deterministic selection” answer above (Weakness 2) for how heads are selected. For a discussion of dynamic or random selection schemes and their comparative evaluation, please refer to our response to Weakness 1.2 to Reviewer 8cpz.
>
> > Q2: Can the author provide some insights on the selected heads? Some other works have shown that different kv_heads play different roles. Are the selected value heads for replacement the relatively redundant ones?
>
> Thank you for your valuable questions. In Appendix C.2, we visualize the **cosine similarities** of all the heads in Layer 1 and 12 of SkipV1Former and standard Transformer. It is shown that SkipV1Former exhibits **lower** inter-head similarity, suggesting some reductions on the redundancy of the heads in standard Transformer.
>
> Regarding the role of heads, prior works have identified previous-token copying heads and induction heads in autoregressive tasks [2]. Specifically, [2] found that there exists a circuit implemented by two sorts of heads:
> - **previous-token copying head**: it finds the previous instance of the current token $A$ and the token that came after $A$ (call it $B$).
> - **induction head**: it “completes the pattern” by predicting $B$ after the current $A$.
>
> This is closely related to our theoretical analysis based on a similar `copying mechanism’ in Section 4. We hypothesize that the skipping heads in SkipV1Former function analogously to previous token copying heads—leveraging layer 1’s complete raw information to facilitate pattern recall—while the remaining heads in layers 2–L perform more similar to induction heads. In effect, SkipV1Former replace some of the less important copying heads in layers 2–L and preserve the more important induction heads. We will include this discussion in the final version of the paper and validate our hypothesis or study other possible interpretation of the selected heads in the future.
>
>
> [2] In-context Learning and Induction Heads
>
> > Q3: What is the batch size for measuring the throughputs in Fig. 14 (b)?
>
> Thank you for your valuable question. The batch size for this experiment is 1, which may lead to underutilization of GPU parallelism and is not representative of steady‑state performance. We have therefore re‑benchmarked inference throughput at batch sizes 8 and 16. As shown in the following table, SkipV1Former performs on par with—and in some cases slightly better than—the baseline (all differences within $\pm 5$%). It seems that the per‑head concatenation of SkipV1Former does not incur measurable overhead as we stated in Appendix C.5, because SkipV1Former has reduced V‑projection matmul FLOPs by approximately 50\%. We expect that the throughput can be further improved for SkipV1Former and leave this to future work. We will include these expanded results and discussions in the final version.
>
> **Table: Inference throughput (tokens/sec) at batch sizes 8 and 16 on an 48G A6000 GPU.**
>
> | Model | Base (B=8) | Skip (B=8) | Base (B=16) | Skip (B=16) |
> |:-----:|:----------:|:----------:|:-----------:|:-----------:|
> | 130M  |    118.6   |    117.9   |    127.0    |    124.8    |
> | 350M  |     59.0   |     61.5   |     53.0    |     54.9    |
> | 1B    |     48.2   |     45.7   |     30.7    |     29.5    |
> | 3B    |     27.3   |     26.2   |     16.2    |     16.7    |
> | 7B    |     13.0   |     13.2   |      —      |      —      |
>
> > Q4: If lowering the compression rate, is SkipV1Former able to operate in a training-free fashion?
>
> Thank you for your valuable question. From our understanding, Minicache and xKV are based on the high similarity between adjacent KV states on the **token/feature dimensions** (or spectral components) of the cache. **Neither method changes the model architecture**. By contrast, SkipV1Former **modifies** the architecture, reducing redundancy **in the head dimension** of multi-head attention (or its variants).  After uptraining, SkipV1Former not only saves memory but also improves overall performance.
>
> Inspired by Minicache and xKV, we think that if a pretrained Transformer already exhibited strong similarity between certain heads in layers 2–L and those in layer 1, one could inject those layer 1 heads directly—effectively a training free SkipV1Former. Prior studies have observed moderate head similarity across layers, but naive head sharing typically causes a marked accuracy drop [3]. Exploring similarity driven, training free methods to turn a standard model into SkipV1Former is an interesting direction. We leave this to future work and we will add a discussion of this potential extension in the final version of the paper.
>
>
> [3] Head-wise Shareable Attention for Large Language Models

---

> > ### Comment · Reviewer_t3Ns · 2025-08-05
> >
> > Thanks for the author's detailed responses. I will maintain my rating.

---

> > > ### Author Response · Authors · 2025-08-05
> > > **Thank you**
> > >
> > > Thank you again for your valuable feedback. We will continue improving our paper.

---

### Official Review · Reviewer_HPMS · 2025-07-02

**Clarity:** 3
**Significance:** 3
**Originality:** 3
**Rating:** 4
**Confidence:** 4

**Summary:**

The paper proposes SkipV1Former, a Transformer variant that reuses half of the Value heads from the first layer in subsequent layers, cutting KV cache by while improving perplexity. It theoretically shows this design restores compressed information and accelerates mesa-optimization, validated by experiments on GPT-2 and LLaMA models, which demonstrate consistent performance gains. An uptraining method converts pre-trained MHAs with extra compute, and SkipV1Former integrates with techniques like GQA for further cache savings. However, it underperforms on non-autoregressive vision tasks, highlighting domain-specific effectiveness.

**Questions:**

Mention in weakness

**Ethical Concerns:**

["NO or VERY MINOR ethics concerns only"]

**Final Justification:**

My concerns have been partially addressed (W2). Regarding Weakness 1, I don't think a 24% reduction in KV cache is worth the performance degradation (from 47.2 to 46.2). Therefore, I will maintain my rating.

**Limitations:**

yes

**Quality:**

3

**Strengths And Weaknesses:**

### Strengths
1. **Innovative Architecture Design**: Introduces SkipV1Former, which strategically reuses half of the Value heads from the first layer in subsequent layers, reducing KV cache while improving model representation. This addresses the trade-off between representation and memory efficiency, outperforming prior skip-connection methods that either sacrificed performance for memory savings or vice versa.
2. **Compatibility with Advanced Methods**: Seamlessly integrates with techniques like Group-Query Attention and Multi-Latent Attention, achieving further KV cache reductions while maintaining or improving performance.

### Weaknesses
1. **Lack comprehensive comparison with other methods**: Though the paper shows the validation loss comparison with related methods (ResFormer) comprehensively in Figure 4, it would be better to report a more comprehensive comparison with other methods (ResFormer) on downstream tasks in Table 1.
2. **Assumption for the theoretical analysis is too strict**: Appendix A attempts to proof that SkipV1Former gains strength in the aspect of "copying mechanism" and in-context-learning. However, the assumption requires no residual connection, which might already have impact on deep layer. Moreover, though authors attempt to proof SkipV1Former benefits deep layers, the assumption stands scenarios with two layers, which might differ from the actual deep layers.

---

> ### Author Rebuttal · Authors · 2025-07-30
>
> Thank you for your careful review and thoughtful questions and comments. Please see our response below.
>
> > Lack comprehensive comparison with other methods: Though the paper shows the validation loss comparison with related methods (ResFormer) comprehensively in Figure 4, it would be better to report a more comprehensive comparison with other methods (ResFormer) on downstream tasks in Table 1.
>
> In addition to Figure 4 and Table 1, we include detailed downstream task comparisons for GPT models with ResFormer and other methods in Tables 5–8 in Appendix C.3. It is shown that SkipV1Former and ResFormer achieve the highest average accuracies across various model scales.
>
> Due to our limited computing resources, we further train a 1 B parameter ResFormer and evaluated it on our downstream benchmarks (see the following table) alongside the 1 B SkipV1Former. Similar to the trend shown on GPT models, ResFormer achieves a marginally higher mean accuracy compared to SkipV1Former. This is reasonable because ResFormer linearly adds all Value heads from layer 1 to subsequent layers without reducing the number of Value heads, thus offering no KV cache savings. On the other hand, SkipV1Former preserves model quality while cutting KV cache by roughly 25 %, demonstrating superior memory efficiency with minimal performance trade off. We will integrate these results in the final version of the paper to give a more comprehensive comparison.
>
> **Table: Test accuracies (%) of 1B scale models on downstream tasks, with overall average.**
>
> |   | $\Delta$ KV  |  ARC-C | ARC-E | BoolQ | Hella | OBQA | PIQA | RTE | SciQ | Winogr | Avg.
> |:-----:|:-----:|:----------:|:----------:|:-----------:|:-----------:|:----------:|:----------:|:----------:|:----------:|:----------:|:----------:|
> | ResFormer-1B | -0%|     20.9   |     47.0   |     56.6    |     32.2    |    18.8    |    68.1    |    53.0    |    77.0    |    52.1    |    47.2    |
> | SkipV1Former-1B | -24%|     19.3   |     44.1   |     59.6    |     31.2    |    17.4    |    66.9    |    52.3    |    73.8    |    50.9    |    46.2    |
>
>
>
> > Assumption for the theoretical analysis is too strict: Appendix A attempts to proof that SkipV1Former gains strength in the aspect of "copying mechanism" and in-context-learning. However, the assumption requires no residual connection, which might already have impact on deep layer.
>
> While our assumption has some limitations, there are some clues supporting that our analysis may apply to standard Transformer.
>
> - It is empirically identified in Ref. [14] that the `linear copying’ mechanism in Assumption 1 appears on the **first‑layer output in standard deep Transformer with residual connections** (See Fig. 2 in Ref. [14] for details).
> - For a simplified one-layer Transformer **with or without** residuals, it is shown respectively in Ref. [46] and Ref. [41] that **both models** implement a step of gradient descent on a least-squares objective when achieving optimality. In other words, it seems that removing residual connections does not impair the representation capability of Transformer for in-context learning.
>
> Together, these findings suggest that omitting residual connections for analytical simplicity does not undermine the core insight we build on in SkipV1Former. We hope these clues can address your concerns and we will include these discussions in the final version of the paper.
>
> > Moreover, though authors attempt to proof SkipV1Former benefits deep layers, the assumption stands scenarios with two layers, which might differ from the actual deep layers.
>
> Thank you for your feedback. Theoretical analyses usually need to start with simple cases to build intuition. Our analysis takes a step towards understanding the benefit of direct injection of Values of layer 1 subsequent layers, and we have provided some discussions on the limitations in Appendix A.2. Extending our analysis to deeper models is an interesting direction, and we will leave this to future works.

---

> > ### Comment · Area_Chair_6yoZ · 2025-08-05
> > **Discussion**
> >
> > Dear Reviewer HPMS,
> >
> > The authors have responded to your concerns. How does their response change your view of the paper? If it does not, please clarify what the authors can do to address your concerns. If it does, please consider adjusting your score based on their response.
> >
> > Your AC

---

> > > ### Author Response · Authors · 2025-08-07
> > > **Thank you**
> > >
> > > Thank you again for your valuable feedback. We will conduct a more comprehensive evaluation on downstream tasks  and continue improving our paper.

---

> > ### Comment · Reviewer_HPMS · 2025-08-06
> >
> > Thanks for your response. My concerns have been partially addressed (W2). Regarding Weakness 1, I don't think a 24% reduction in KV cache is worth the performance degradation (from 47.2 to 46.2). Therefore, I will maintain my rating.

---

### Note · Authors · 2025-08-13

We sincerely thank the Area Chair and all reviewers for their careful reading and constructive feedback.

**Our contributions**:
In this paper, we propose **SkipV1Former**, a Transformer variant that reuses half of the first layer’s Value heads in every subsequent layer, reducing the overall KV cache by ~25% while improving model performance. We also provide a theoretical analysis explaining why the skip connection helps and demonstrate that SkipV1Former can be combined with other advanced KV-saving methods.

**Reviewers ' main positive takeaways**:
Reviewers converged on three main strengths: 1) the novelty and practicality of the architectural idea that jointly addresses representation and KV memory; 2) a useful blend of theory and experiments—Section 4 and Appendix A provide intuition and analysis that reviewers found insightful; and 3) versatility and the ability of SkipV1Former to be seamlessly integrated with advanced KV-saving methods such as GQA, MLA, YOCO.

**Rebuttal work-what we did and why**:
During the rebuttal, we focused on reviewers’ concerns about performance improvements, how and why KV is saved, and ablations. 1) Model performance: We clarify that SkipV1Former delivers **both higher accuracy and ~25% KV-cache reduction across scales**. 2) KV saving mechanism: We contrast our decoding with a standard Transformer and highlight our theory showing why a skip connection from layer 1 helps under a mesa-optimization perspective. 3) Abalations: We vary the skip-head ratio and the head-insertion strategies on 1B models. Results indicate our default settings offer a strong accuracy–memory tradeoff with good systems compatibility; larger skip-head ratios are also attractive in memory-constrained environments.

**Summary of additional experiments**:
We add: 1) seed-robustness checks for 1B pretraining; 2) GQA-4 integration on TinyLLaMA-1.1B; 3) downstream evaluation of the 1B ResFormer; 4) ablations on skip-head ratios (25/50/75) and head-insertion strategies; 5) ablations on the depth of Value injection. These results further support our claims.

**Current status of reviews**:
Positive reviewers (HPMS, t3Ns) have maintained their scores, and reviewers who raised concerns (8cpz, oBgN) indicated that our rebuttal mostly addressed their issues and would influence their final scoring.

We appreciate the reviewers’ detailed critiques—their suggestions materially strengthened the paper—and we will incorporate all clarifications in the final submission.

---

### Decision · Program_Chairs · 2025-09-17

**Decision:**

Accept (poster)

**Comment:**

All reviewers agreed this paper should be accepted: it includes a nice mathematical analysis motivating their proposed solution, it modularly combines with other efficient methods, and has useful experiments. The main concerns were around (a) limited improvements: e.g., reviewer oBgN pointed out the marginal improvements in accuracy. The reviewers responded that the method provides both improvements in accuracy and KV-cache reduction, pointing out that other KV-cache-saving methods, e.g., SVD-based compression methods, require additional computation. This answers the concern. Another concern was around (b) comparison with other methods: e.g., reviewer HPMS wanted to see downstream task comparisons. These were provided by the authors, addressing this concern as well. Given these and other responses by authors to reviewer concerns, the paper is a clear accept. Authors please remember to include all reviewer requests into the camera-ready paper. Thank you!